# Continuous Soft Actor-Critic: An Off-Policy Learning Method Robust to Time Discretization

**Huimin Han**
Zhongtai Securities Institute for Financial Studies
Shandong University
Jinan, 250100 P. R. China
`hanhuiminhhm@mail.sdu.edu.cn`

**Shaolin Ji** *
Zhongtai Securities Institute for Financial Studies
Shandong University
Jinan, 250100 P. R. China
`jsl@sdu.edu.cn`

## Abstract

Many *Deep Reinforcement Learning* (DRL) algorithms are sensitive to time discretization, which reduces their performance in real-world scenarios. We propose Continuous Soft Actor-Critic, an off-policy actor-critic DRL algorithm in continuous time and space. It is robust to environment time discretization. We also extend the framework to multi-agent scenarios. This *Multi-Agent Reinforcement Learning* (MARL) algorithm is suitable for both competitive and cooperative settings. Policy evaluation employs stochastic control theory, with loss functions derived from martingale orthogonality conditions. We establish scaling principles for hyperparameters of the algorithm as the environment time discretization $\delta t$ changes ($\delta t \to 0$). We provide theoretical proofs for the relevant theorems. To validate the algorithm's effectiveness, we conduct comparative experiments between the proposed algorithm and other mainstream methods across multiple tasks in *Virtual Multi-Agent System* (VMAS). Experimental results demonstrate that the proposed algorithm achieves robust performance across various environments with different time discretization parameter settings, outperforming other methods.

## 1 Introduction

Recently, reinforcement learning algorithms such as *Proximal Policy Optimization* (PPO, Schulman et al. [2017]), *Soft Actor-Critic* (SAC, Haarnoja et al. [2018a]), and *Deep Deterministic Policy Gradient* (DDPG, Lillicrap et al. [2016]) have demonstrated remarkable success in domains such as large language models, robotics, autonomous driving, and so on. However, there remains little research on the continuous-time learning algorithms for stochastic environments and multi-agent reinforcement learning. And the achievements of deep reinforcement learning studies under the discrete-time frameworks may lack robustness to time discretization. Experimental studies by Henderson et al. [2018] and Tallec et al. [2019] verify that many DRL algorithms developed under discrete-time frameworks lack robustness to the hyperparameters, particularly the time step. Furthermore, Tallec et al. [2019] formally prove that $Q$-learning does not exist in continuous time, though their analysis is conducted under deterministic environment. These findings raise the following question:

---

*Corresponding author

39th Conference on Neural Information Processing Systems (NeurIPS 2025).

- Do algorithms such as DQN (Mnih et al. [2015]) and SAC collapse when the time discretization step $\delta t \to 0$ in stochastic environments?
- Can we propose a robust algorithm for time discretization under stochastic environments?
- How should we adjust the hyperparameters of the algorithm when the time step $\delta t$ changes?

## 1.1 Original contributions

Aiming to answer the aforementioned questions, we establish a theoretical framework for stochastic continuous reinforcement learning utilizing stochastic control theory. The contributions of this research are listed as follows:

(i) This paper presents a comprehensive analysis of continuous-time reinforcement learning algorithms in stochastic environments. We investigate the optimality of value function approximation, the impact of time discretization, hyperparameter settings, and algorithm implementation. To the best of our knowledge, this work also pioneers the first finite multi-agent actor-critic algorithm designed for continuous-time settings, offering the analysis of finite multi-agent systems in stochastic continuous-time environments.

(ii) In deterministic environments such as Tallec et al. [2019] and Doya [2000], alterations to the time discretization parameter ($\delta t$) introduce fundamental inconsistencies. In stochastic environments, both the configuration of value function approximation and time discretization changes introduce significant challenges. The *mean squared temporal difference error* (MSTDE) used in *policy evaluation* (PE) cannot be applied to stochastic continuous setting. This invalidates popular RL algorithms such as DQN and SAC fail to approximate true value functions in such environments. To address this, we use a novel PE method grounded in martingale theory.

(iii) We derive the hyperparameter scaling laws for our proposed algorithm under stochastic continuous settings as the time step $\delta t$ changes. This is different from those proposed by Tallec et al. [2019] under the ordinary differential equation framework.

(iv) The MARL algorithm developed in this research is designed for finite-agent systems and adapts to both competitive and cooperative scenarios. We provide experimental implementation of the algorithm in stochastic continuous-time settings. The implementation leverages BenchMARL (Bettini et al. [2024]) to compare with MARL baselines, including MASAC, MAPPO (Yu et al. [2022]), MADDPG (Lowe et al. [2017]), IQL (Tan [1993]), and QMIX (Rashid et al. [2020]) across various environments. Empirical evidence demonstrates that performance of algorithms such as IQL significantly degrades as the time step $\delta t$ decreases, aligning with theoretical predictions. Experimental results preliminarily indicate that our proposed Continuous (Multi-Agent) Soft Actor-Critic (abbreviated as CSAC and CMASAC) outperforms other methods when $\delta t$ is small, confirming its robustness to time discretization.

## 1.2 Related work

**Continuous-time reinforcement learning** Baird [1994] and Doya [2000] studied algorithms in the limit of discrete time step $\delta t$ approaching zero, from discrete-time and continuous-time perspectives, respectively. Tallec et al. [2019] formally proved that $Q$-learning cannot exist in continuous-time deterministic environments. Consequently, methods based on $Q$-learning such as DQN and DDPG also fail in continuous-time settings. Tallec et al. [2019] further provided experimental evidence that DQN and DDPG lack robustness to variations in time discretization. Doya [2000] and Tallec et al. [2019] provide studies of algorithms with continuous-time limits under deterministic environments. From a stochastic perspective, Jia and Zhou [2022a] and Jia and Zhou [2022b] investigate the Policy Evaluation problem under the framework of stochastic optimal control theory. However, their work focuses on the conceptual aspects of on-policy frameworks. There has been no substantial exploration of practical off-policy algorithms, impacts of time discretization scales, hyperparameter tuning, or implementation considerations. Through our design, we ensure the feasibility of gradient backpropagation during policy updates and propose a practical off-policy algorithm robust to time discretization.

**Soft Actor-Critic** Haarnoja et al. [2018a] and Haarnoja et al. [2018b] developed an off-policy actor-critic DRL algorithm based on the maximum entropy framework. This approach has become

a well-performing RL algorithm on a range of continuous control tasks. In this research, we also consider continuous states and actions, with the actor aiming to maximize the Shannon entropy.

**Multi-agent reinforcement learning**   Multi-agent reinforcement learning has achieved practical successes in autonomous driving (Mirowski et al. [2017]), AlphaGo (Silver et al. [2016]), and StarCraft (Vinyals et al. [2019]), but with few results in continuous-time settings. Wang and Zhou [2020] establish a continuous framework for entropy-regularized RL. For large-scale agent systems, mean-field game approaches such as those proposed by Guo et al. [2022], Guo et al. [2024] have been explored. For finite agent systems, algorithms such as Yu et al. [2022], Lowe et al. [2017] are compared in this work.

## 2   Preliminaries

We first briefly introduce frameworks for continuous-time reinforcement learning problems and analyze their properties within these frameworks. Time discretization occurs when implementing these algorithms.

### 2.1   Framework

Let $(\Omega, \mathcal{F}, \mathbb{P}; \{\mathcal{F}_t\}_{t \geq 0})$ be a filtered probability space, in which denote a standard n-dimensional Brownian motion $W = \{W_t, t \geq 0\}$. $\{\mathcal{F}_t^W\}_{t \geq 0}$ denote the natural filtration generated by $W$, and $\mathcal{P}(U)$ denote the set of probability measures taking values on the action space $U$. [2] We denote $Z_t$ by random variable that is uniformly distributed on $[0, 1]$, independent of $W$, and $Z_t, 0 \leq t \leq s$ are mutually independent, then $\mathcal{F}_s = \mathcal{F}_s^W \vee \sigma(Z_t, 0 \leq t \leq s)$. The admissible control $u^\pi = \{u_t^\pi, 0 \leq t \leq T\}$ is $\{\mathcal{F}_t\}_{t \geq 0}$-progressively measurable process representing the actions generated by agent's policy $\pi(\cdot | t, x) \in \mathcal{P}(U)$. The agent aims to control the stochastic dynamical system:

$$\begin{cases} dX^\pi(t) = b\left(t, X^\pi(t), u^\pi(t)\right) dt + \bar{\sigma}\left(t, X^\pi(t), u^\pi(t)\right) dW(t), \\ X^\pi(0) = x_0 \in \mathbb{R}^n, \quad t \geq 0 \end{cases} \tag{1}$$

to maximize the entropy-regularized expected cumulative reward:

$$J(0, x_0; \pi) = \mathbb{E}_{\mathbb{P}} \left[ \int_0^T e^{-\beta t} \left[ r(t, X_t^\pi, u_t^\pi) - \lambda \ln \pi(u_t^\pi | t, X_t^\pi) \right] dt + e^{-\beta T} h(X_T^\pi) | \mathcal{F}_0 \right]. \tag{2}$$

The optimal value function $V(t, x)$ for state $x$ at time $t$ is defined by

$$V(t, x) = \sup_{\pi \in \mathcal{P}(U)} J(t, x; \pi).$$

### 2.2   Multi-agent framework

We denote $U$ and $V$ by the action spaces of two agents respectively, $u^\pi(t)$, $v^\pi(t)$ representing actions generated by policy $\pi^u$, $\pi^v$ respectively. Analogous to the single-agent setting, the corresponding system dynamics and value functions for agents with their policies $\pi = (\pi^u, \pi^v)$, can be formulated as:

$$\begin{cases} dX^\pi(t) = b\left(t, X^\pi(t), u^\pi(t), v^\pi(t)\right) dt + \bar{\sigma}\left(t, X^\pi(t), u^\pi(t), v^\pi(t)\right) dW(t), \\ X^\pi(0) = x_0 \in \mathbb{R}^n, \quad t \geq 0 \end{cases} \tag{3}$$

---

[2]In this paper, policies are modeled as parametric distributions (Gaussian policies). To align with conventional notation (e.g., Sutton and Barto [2018]), we do not strictly differentiate between density functions, probability measures, and policies, instead uniformly representing them by the notation $\pi \in \mathcal{P}(U)$.

and
$$V_1(t,x) = \sup_{\pi^v \in \mathcal{P}(V)} J_1(t,x;\bar{\pi}^u, \pi^v) =$$

$$\sup_{\pi^v \in \mathcal{P}(V)} \mathbb{E}_{\mathbb{P}} \left[ \int_t^T e^{-\beta(s-t)} \left[ r_1\left(s, X_s^\pi, u_s^{\bar{\pi}}, v_s^\pi\right) - \lambda_1 \ln \pi^v(v^\pi|s, X_s^\pi) \right] ds + e^{-\beta(T-t)} h_1(X_T^\pi)|\mathcal{F}_t \right],$$

$$V_2(t,x) = \sup_{\pi^u \in \mathcal{P}(U)} J_2(t,x;\pi^u, \bar{\pi}^v) =$$

$$\sup_{\pi^u \in \mathcal{P}(U)} \mathbb{E}_{\mathbb{P}} \left[ \int_t^T e^{-\beta(s-t)} \left[ r_2\left(s, X_s^\pi, u_s^\pi, v_s^{\bar{\pi}}\right) - \lambda_2 \ln \pi^u(u^\pi|s, X_s^\pi) \right] ds + e^{-\beta(T-t)} h_2(X_T^\pi)|\mathcal{F}_t \right],$$
$$(4)$$

where $(t,x) \in [0,T] \times \mathbb{R}^n$, $(\bar{\pi}^u, \bar{\pi}^v)$ are fixed policies, and the optimal policies $\pi^*$ satisfy:
$$\pi^{*u} = \arg\max_{\pi^u \in \mathcal{P}(U)} J_2\left(t,x;\pi^u, \pi^{*v}\right), \quad \pi^{*v} = \arg\max_{\pi^v \in \mathcal{P}(V)} J_1\left(t,x;\pi^{*u}, \pi^v\right). \tag{5}$$

We assume that problems (1)-(2) and (3)-(4) remain well-posed throughout the analysis. [3] For simplicity, this paper focuses on a two-agent scenario. We emphasize that all results can be naturally extended to finite multi-agent systems.

## 3  Main results

### 3.1  Policy evaluation

**Mean squared temporal difference error**  In discrete-time settings, algorithms like DQN, DDPG, and SAC apply Bellman's principle of optimality to estimate state-action value function $Q(x,u)$, defining their loss functions with *mean-square TD error*. As for continuous settings, Doya [2000] also adopts this method for state value function $J(x)$. However, Jia and Zhou [2022a] argue that in stochastic environments, this method cannot guarantee convergence to the value function. We outline a brief description about this issue below.

Let us recall Doya's TD Algorithm. The deterministic system satisfies:
$$J(t,x_t) = \int_t^{t'} \mathbf{r}(s,x_s)ds + J(t',x_{t'}), \quad t \in [0,T], \tag{6}$$

where $\mathbf{r}$ can encapsulates terms such as reward, discount, and regularization. Using parameterized functions $J^\theta$, Doya aims at minimizing the *mean-square TD error*:
$$\dot{J}_t = \frac{d}{dt} J(t,x_t), \text{MSTDE}(\theta) = \frac{1}{2} \int_0^T [\dot{J}_t(t,x_t) + \mathbf{r}(t,x_t)]^2 dt = \frac{1}{2} \int_0^T \left( \dot{J}_t^\theta + \mathbf{r}(t,x_t) \right)^2 dt. \tag{7}$$

For stochastic system (1), (3), for any fixed $(t,x) \in [0,T] \times \mathbb{R}^n$, define process:
$$M_s := J(s,X_s) + \int_t^s \mathbf{r}(s',X_{s'})\,ds', \quad s \in [t,T]. \tag{8}$$

$\{M_s, t \le s \le T\}$ is a square-integrable martingale. Regarding uniform discrete time intervals $\delta t$, when $\delta t \to 0$, the discrete-time MSTDE becomes
$$\text{MSTDE} := \frac{1}{2} \mathbb{E} \left[ \sum_{i=0}^{K-1} \left( \frac{J(t_{i+1}, X_{t_{i+1}}) - J(t_i, X_{t_i})}{t_{i+1} - t_i} + \mathbf{r}(t_i, X_{t_i}) \right)^2 \cdot \delta t \right]$$

$$= \frac{1}{2\delta t} \mathbb{E} \left[ \sum_{i=0}^{K-1} \left| J(t_{i+1}, X_{t_{i+1}}) - J(t_i, X_{t_i}) + \int_{t_i}^{t_{i+1}} \mathbf{r}_s ds + \mathcal{O}((\delta t)^2) \right|^2 \right] \approx \frac{1}{2\delta t} \langle M \rangle_T \ne 0$$
$$(9)$$

where $\langle M \rangle_T$ denotes the quadratic variation of the martingale $M$. From equation (9), it can be observed that the MSTDE corresponding to the true value function $J$ is not zero. Therefore, minimizing MSTDE to learn parameterized function $J^\theta$ cannot guarantee convergence to value function $J$ in continuous-time and space environments. The martingale property of the value function motivates a novel approach to policy evaluation.

---

[3]This point is guaranteed by Assumption 2 and Definition 1 provided in the Appendix B.

**Martingale orthogonality condition**   To ensure that the learning process eventually converges to the value function, we impose constraints on the parameterized function $J^\theta$ to preserve the martingale property. Jia and Zhou [2022a] propose the following proposition.

**Proposition 1.** *A process $M \in L^2_{\mathcal{F}}([0,T])$ is a martingale if and only if*

$$\mathbb{E}\left[\int_0^T \xi_t dM_t\right] = 0, \quad \textit{for any} \quad \xi_t \in L^2_{\mathcal{F}}([0,T], M). \tag{10}$$

We assume all value functions $J$ and their approximators $J^\theta$ discussed in this work satisfy the following assumptions.

**Assumption 1.** *For all $\theta \in \Theta, J, J^\theta \in C^{1,2}([0,T) \times \mathbb{R}^n) \cap C([0,T] \times \mathbb{R}^n)$ and satisfies the polynomial growth condition in x. Moreover, $J^\theta(t,x)$ is a smooth function in $\theta$. $\frac{\partial J^\theta}{\partial \theta}, \frac{\partial^2 J^\theta}{\partial \theta^2} \in C^{1,2}([0,T) \times \mathbb{R}^n) \cap C([0,T] \times \mathbb{R}^n)$ satisfying the polynomial growth condition in x.*

**Theorem 1.** *A function is the value function associated with the policy $\pi$ if it satisfies terminal condition $J(T,x;\pi) = h(x)$, and for any given $(t,x) \in [0,T] \times \mathbb{R}^n$ and admissible policy $\tilde{\pi} \in \mathcal{P}(U)$, define*

$$M_s := e^{-\beta s} J(s, X_s^{\tilde{\pi}}; \pi) + \int_t^s e^{-\beta s'} \left[ \left( r(s', X_{s'}^{\tilde{\pi}}, u_{s'}^{\tilde{\pi}}) - \lambda \ln \pi(u^{\tilde{\pi}}|s', X_{s'}^{\tilde{\pi}}) \right) \right] ds' \tag{11}$$

*is an $(\mathcal{F}_s, \mathbb{P})$-martingale on $[t,T]$.*

According to Proposition 1 and Theorem 1, the approximating function $J^\theta$ is the value function associated with policy $\pi$ when the martingale orthogonality condition:

$$\mathbb{E}_{\mathbb{P}} \int_0^T \xi_t \left[ dJ^\theta(t, X_t^{\tilde{\pi}}; \pi) + r(t, X_t^{\tilde{\pi}}, u_t^{\tilde{\pi}})dt - \lambda \ln \pi(u^{\tilde{\pi}}|t, X_t^{\tilde{\pi}})dt - \beta J^\theta(t, X_t^{\tilde{\pi}}; \pi)dt \right] = 0 \tag{12}$$

for any $\xi_t \in L^2_{\mathcal{F}}[0,T]$ holds.

In this work, we propose an off-policy algorithm to learn the value function by imposing the constraint:

$$\mathbb{E}_{\mathbb{P}} \int_0^T \xi_t \left[ dJ^\theta(t, X_t^{\tilde{\pi}}; \pi^\phi) + r(t, X_t^{\tilde{\pi}}, u_t^{\tilde{\pi}})dt - \lambda \ln \pi^\phi(u_t^{\tilde{\pi}}|t, X_t^{\tilde{\pi}})dt - \beta J^\theta(t, X_t^{\tilde{\pi}}; \pi^\phi)dt \right] = 0. \tag{13}$$

Here, $\tilde{\pi}$ denotes the behavior policy, $\pi^\phi$ denotes the approximated policy parameterized by neural network. This implies that we can learn the value function $J$ of a given target policy $\pi$ based on data generated by a different admissible policy $\tilde{\pi}$. We emphasize that the state transitions (1) and rewards $r$ are inherent properties of the environment, determined only by the current state and action, and independent of the policy $\pi$. Similar to the framework in Haarnoja et al. [2018a], where policies are modeled as Gaussian distributions with entropy regularization, all policies cover the same action space, and reparameterized sampling is employed in the policy gradient step, making importance sampling weights unnecessary. [4]

**Martingale orthogonality condition for multi-agent systems**   For multi-agent systems, the martingale orthogonality conditions also hold. By generalizing Theorem 1, where we denote the policy of agent 2 as $\bar{\pi}^u$, the following theorem guarantees the policy evaluation for agent 1.

**Theorem 2.** *A function $J_1(\cdot, \cdot; \bar{\pi}^u, \pi^v)$ is the value function associated with the policy $\pi = (\bar{\pi}^u, \pi^v)$ if it satisfies the terminal condition $J_1(T, x; \bar{\pi}^u, \pi^v) = h_1(x)$, and for any fixed $(t,x) \in [0,T] \times \mathbb{R}^n$ and admissible policies $(\bar{\pi}^u, \tilde{\pi}^v)$, define*

$$
\begin{aligned}
M_s^1 := & e^{-\beta s} J_1(s, X_s^{\tilde{\pi}}) + \int_t^s \left[ e^{-\beta s'} \left( \int_U r_1(s', X_{s'}^{\tilde{\pi}}, u_{s'}^{\bar{\pi}}, v_{s'}^{\tilde{\pi}}) \bar{\pi}^u(u^{\bar{\pi}}|s', X_{s'}^{\tilde{\pi}})du \right. \right. \\
& \left. \left. - \lambda_1 \ln \pi^v(v^{\tilde{\pi}}|s', X_{s'}^{\tilde{\pi}}) \right) \right] ds'
\end{aligned}
\tag{14}
$$

*is an $(\mathcal{F}_s, \mathbb{P})$-martingale on $[t,T]$.*

---

[4] Proof of Theorem 1 and a more detailed discussion can be found in the Appendix B.1 and Appendix C.

By integrating Theorem 2 with the single-agent analysis presented earlier, we derive a critic network update rule for multi-agent reinforcement learning analogous to (13):

$$\mathbb{E}_{\mathbb{P}} \int_0^T \xi_t^1 \left[ dJ_1^{\theta_1}(t, X_t^{\tilde{\pi}}; \bar{\pi}^u, \pi_\phi^v) + r_1(t, X_t^{\tilde{\pi}}, \bar{u}_t^\pi, v_t^{\tilde{\pi}})dt - \lambda_1 \ln \pi_\phi^v(v^{\tilde{\pi}}|t, X_t^{\tilde{\pi}})dt \right.$$
$$\left. - \beta J_1^{\theta_1}(t, X_t^{\tilde{\pi}}; \bar{\pi}^u, \pi_\phi^v)dt \right] = 0. \tag{15}$$

The evaluation of $J_2$ also obeys the above theorem and condition.

## 3.2 Policy gradient

The following theorems are proposed to characterize policy gradients in both single-agent and multi-agent reinforcement learning. The related proofs are provided in the Appendix B. These theorems extend the policy gradient theorem in Jia and Zhou [2022b] to broader settings.

**Theorem 3.** *Consider an admissible parameterized policy $\pi^\phi$ within the dynamical system (1), for any $(t, x) \in [0, T] \times \mathbb{R}^n$, its policy gradient $g(t, x; \phi) = \frac{\partial}{\partial \phi} J\left(t, x; \pi^\phi\right)$ admits the following representation:*

$$g(t, x; \phi) = \mathbb{E}_{\mathbb{P}} \left[ \int_t^T e^{-\beta(s-t)} \left\{ \frac{\partial}{\partial \phi} \ln \pi^\phi(u_s|s, X_s^\pi) \left( dJ\left(s, X_s^\pi; \pi^\phi\right) + [r\left(s, X_s^\pi, u_s\right) \right. \right. \right.$$
$$\left. \left. \left. + \lambda \ln \pi^\phi\left(u_s|s, X_s^\pi\right) - \beta J\left(s, X_s^\pi; \pi^\phi\right)] ds \right) - \lambda \frac{\partial}{\partial \phi} \ln \pi^\phi\left(u_s|s, X_s^\pi\right) ds \right\} \mid X_t^\pi = x \right]. \tag{16}$$

**Theorem 4.** *Consider an admissible parameterized policy $\pi^\phi = (\bar{\pi}_{\phi_2}^u, \pi_{\phi_1}^v)$ within the dynamical system (3), for any $(t, x) \in [0, T] \times \mathbb{R}^n$, its policy gradient $g_1(t, x; \bar{\phi}_2, \phi_1) := \frac{\partial J_1(t, x; \bar{\pi}_{\phi_2}^u, \pi_{\phi_1}^v)}{\partial \phi_1}$ admits the following representation:*

$$g_1(t, x; \bar{\phi}_2, \phi_1)$$
$$= \mathbb{E}_{\mathbb{P}} \left[ \int_t^T e^{-\beta(s-t)} \left\{ \frac{\partial}{\partial \phi_1} \ln \pi_{\phi_1}^v(v_s|s, X_s^\pi) \left( dJ_1\left(s, X_s^\pi; \bar{\phi}_2, \phi_1\right) + [r\left(s, X_s^\pi, u_s, v_s\right) \right. \right. \right.$$
$$\left. - \lambda_1 \ln \pi^v\left(v_s|s, X_s^\pi\right) - \beta J\left(s, X_s^\pi; \bar{\phi}_2, \phi_1\right)] ds \right)$$
$$\left. - \lambda_1 \frac{\partial}{\partial \phi} \ln \pi^v\left(v_s|s, X_s^\pi\right) ds \right\} \mid X_t^\pi = x \right], \quad \forall (t, x) \in [0, T] \times \mathbb{R}^d. \tag{17}$$

The policy gradient of another agent $g_2(t, x; \phi_2, \bar{\phi}_1) := \frac{\partial J_1(t, x; \pi_{\phi_2}^u, \bar{\pi}_{\phi_1}^v)}{\partial \phi_2}$ has same representation.

## 3.3 Continuous Soft Actor-Critic

**Continuous Soft Actor-Critic** The entropy term in (2) enhances the exploration capability of policy $\pi$. Hyperparameters (e.g., temperature $\lambda$) and soft update techniques follow the implementation of Haarnoja et al. [2018b]. To stabilize training, we employ a separate function approximator for the soft value $J(t, x_t)$, which minimizes the martingale orthogonality conditions through gradient descent. We integrate key techniques for automatic adaptation of temperature parameters $\lambda$ via dual gradient descent, as proposed by Haarnoja et al. [2018b]. We choose $\xi_t = \frac{\partial J^\theta(t, X_t)}{\partial \theta}$ and we optimize the temperature parameter $\lambda$ via:

$$\lambda_t^* = \arg\min_{\lambda_t} \mathbb{E}_{\mathbb{P}}[-\lambda_t \ln \pi(u|t, x) - \lambda_t \bar{\mathcal{H}}], \tag{18}$$

where the target entropy $\bar{\mathcal{H}}$ is typically task-specific. In practice, we solve (18) using stochastic gradient descent. Algorithm 1 outlines the entire procedure.

**Continuous Multi-Agent Soft Actor-Critic** The algorithmic workflow of the multi-agent system aligns with the single-agent case, where the function approximator $J$, reward $r$, and entropy term are replaced by their multi-agent counterparts. We adopt the *Centralized Training with Decentralized Execution* (CTDE) framework. Coordination between the two agents is achieved through alternating updates. The complete procedure is detailed in Algorithm 2 in Appendix A.

---

**Algorithm 1** Continuous Soft Actor-Critic Algorithm

---

**Inputs:** time step $\delta t$, number of epochs $N$, number of mesh grids $K$, number of gradient step $L$, batch size $I$, initial learning rates $\alpha_\theta, \alpha_\phi$, initial $\theta, \phi, \theta_{\text{target}}$, discount factor $\beta$, and temperature parameter $\lambda$. $J^\theta(\cdot, \cdot)$ defining functional form of the value function, $\pi_\phi(\cdot \mid \cdot)$ defining functional form of the policy, $\mathcal{D}$ defining buffer of transitions, $\boldsymbol{opt}_J, \boldsymbol{opt}_\pi$ defining optimizer.

**Interactive program:** an environment simulator $(x', r) = \text{Environment}_{\delta t}(t, x, u)$ that takes current time-state pair $(t, x)$ and action $u$ as inputs and generates state $x'$ at time $t + \delta t$ and the instantaneous reward $r$ at time $t$.

**Learning procedure:**

**for** $j = 1$ **to** $N$ **do**

    Initialize $k = 0$. Observe the initial state $x_0$ and store $x_{t_k} \leftarrow x_0$.

    **while** $k < K$ **do**

        Generate action $u_{t_k} \sim \pi_\phi(\cdot \mid t_k, x_{t_k})$.

        Apply $u_{t_k}$ to the environment simulator $(x, r) = \text{Environment}_{\delta t}(t_k, x_{t_k}, u_{t_k})$, and observe the output new state $x$ and reward $r$.

        Store $x_{t_{k+1}} \leftarrow x$ and $r_{t_k} \leftarrow r$ in $\mathcal{D}$, $\mathcal{D} = \mathcal{D} \cup (x_{t_k}, u_{t_k}, r_{t_k}, x_{t_{k+1}}, d_{t_{k+1}})$, $d$ is the episode termination signal. Update $k \leftarrow k + 1$.

    **end while**

    **for** $l = 1$ **to** $L$ **do**

        Sampled a batch of $I$ transitions $(x_{t_i}, x_{t_{i+1}}, u_{t_i}, r_{t_i})$ from $\mathcal{D}$ and a batch of $u_{t_i}^\phi$ from $\pi^\phi$. Compute

$$
\begin{aligned}
\delta M =& J^{\theta_{\text{target}}}\left(t_{i+1}, x_{t_{i+1}}\right) - J^\theta\left(t_i, x_{t_i}\right) + r_{t_i}\delta t - \lambda \ln\left(\pi_\phi\left(u_{t_i} \mid t_i, x_{t_i}\right)\right)\delta t \\
& - \beta J^\theta\left(t_i, x_{t_i}\right)\delta t, \\
\Delta\theta =& \frac{1}{I}\sum_{i=0}^{I-1}\frac{\partial_\theta J^\theta(t_i, x_{t_i})\delta M}{\delta t}, \\
\Delta\phi =& \frac{1}{I}\sum_{i=0}^{I-1}\frac{\left[\partial_\phi \ln \pi_\phi\left(u_{t_i}^\phi \mid t_i, x_{t_i}\right)\right]\delta M - \lambda\partial_\phi \ln\left(\pi_\phi\left(u_{t_i}^\phi \mid t_i, x_{t_i}\right)\right)\delta t}{\delta t}.
\end{aligned}
\tag{19}
$$

        Update $\theta$ (policy evaluation) with $\boldsymbol{opt}_J$, learning rate $\alpha_\theta\delta t$, and $\Delta\theta$.

        Update $\phi$ (policy gradient) with $\boldsymbol{opt}_\pi$, learning rate $\alpha_\phi\delta t$ and $\Delta\phi$.

        Adjust temperature $\lambda$. Update $\theta_{\text{target}} \leftarrow \tau\theta + (1-\tau)\theta_{\text{target}}$.

    **end for**

**end for**

---

### 3.4 Hyperparameter scaling

The optimality of value functions in our continuous-time algorithm is well established. Here, we analyze the impact of time step $\delta t$ on parameter updates. Time-dependent scaling laws for returns, discount factors, learning rates, and temperatures are derived from discrete-time formulations ($\delta t = 1$).

**Learning rate scaling**  As proven in Tallec et al. [2019], the state-action value function $Q(x, u)$ and state value function $V(x)$ after time discrete satisfy: $Q_{\delta t}^\pi(x, u) = V_{\delta t}^\pi(x) + \mathcal{O}(\delta t)$. In learning framework, the advantage function satisfy: $A_{\delta t}^\psi(x, u) = Q_{\delta t}(x, u) - V_{\delta t}^\theta(x) = \mathcal{O}(\delta t)$. Consequently, $Q$-learning or $A$-learning gradients scale as $\mathcal{O}(\delta t)$, causing vanishing gradients during backpropagation. Empirical validation is provided in Table 6, which displays MADDPG gradients across four update steps. To address this, Tallec et al. [2019] redefine the advantage function as $A_{\delta t}^\psi(x, u) = \frac{Q_{\delta t}(x, u) - V_{\delta t}^\theta(x)}{\delta t} = \mathcal{O}(1)$ and scaling the learning rate.

Extending this insight to stochastic continuous environments, the martingale difference term also satisfies $\delta M_t^\theta = \mathcal{O}(\delta t)$. To stabilize gradient magnitudes, we normalize it as $\delta M_t^\theta/\delta t$, while preserving the martingale structure of $J^\theta$. Table 7 validates this normalization strategy by comparing gradients under scaled versus unscaled configurations across three update steps in the proposed algorithm.

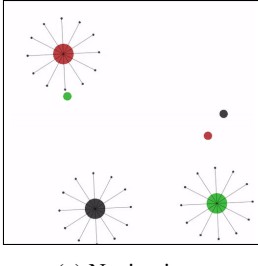
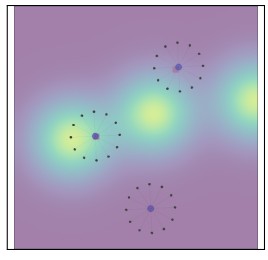

(a) Navigation.  (b) Sampling.

Figure 1: Two VMAS multi-robot control tasks used in the experiments.

We analyze the algorithm in a continuous-time framework with temporal discretization during implementation. To ensure that discrete-time parameter trajectories converge to well-defined continuous-time limits, learning rate scaling is essential. This is formalized in the following theorem.

**Theorem 5.** *Let $(x_t, u_t)$ be some exploration trajectory under time discretization. Set the learning rates to $\eta_{\delta t}^{\theta} = \alpha_\theta \delta t^\beta$ and $\eta_{\delta t}^{\phi} = \alpha_\phi \delta t^\beta$ for some $\beta > 0$, and learn the parameters $\theta$ and $\phi$ by iterating (19) along the trajectory $(x_t, u_t)$. Then, when $t > 0$:*

*(i) If $\beta = 1$ the discrete parameter trajectories converge to continuous parameter trajectories;*

*(ii) If $\beta > 1$ the parameters stay at their initial values;*

*(iii) If $\beta < 1$, the parameters can reach infinity.*

**Other hyperparameter scalings**  (i) Discount factor: By comparing our framework with the continuous *Markov Decision Process* (MDP) formulation in Tallec et al. [2019], we derive the relationship $\gamma = e^{-\beta}$. Then from the parameter update rule in Equation (19), the discrete-time discount factor $e_{\delta t}^{-\beta}$ under variable time step scaling becomes: $e_{\delta t}^{-\beta} = e^{-\beta \delta t} = \gamma_{\delta t}$. (ii) Temperature: Equation (19) demonstrates linear scaling of the temperature parameter with the discretization interval: $\lambda_{\delta t} = \lambda \cdot \delta t$. (iii) Reward: The reward term scales proportionally to the time step via formula (19): $r_{\delta t} = r \cdot \delta t$.

## 4 Experiments

**Tasks**  We conducted experiments using multiple tasks in the VMAS simulator (Bettini et al. [2022]). The visual representations of the *Navigation* and *Sampling* tasks are illustrated in Figure 1.

**Implementations**  Following Bettini et al. [2024], we employ their network architectures, default hyperparameters, and other configurations in our experiments. Descriptions of tasks, random seeds, network architectures, optimizer settings and other implementation details are all documented in the Appendix E. [5] For fair comparison, the proposed algorithm in this work utilizes the common hyperparameters tuned in Bettini et al. [2024], which may not reflect its optimal performance, yet remains valid for robustness verification.

**Results**  Here, we employ the simulation environments *navigation* and *sampling* (validated in Bou et al. [2024]), along with multi-agent reinforcement learning algorithms including MAPPO, MASAC, and MADDPG (which demonstrated superior performance in Bettini et al. [2024]), as well as $Q$-learning-based approaches QMIX and IQL, to conduct a comparative analysis of these algorithms.

(i) As shown in Table 4, when $\delta t$ decreases from 0.1 to 0.01, the performance of popular algorithms gradually declines. We selected two representative algorithms—MADDPG, which demonstrated the best performance in experiments from Bettini et al. [2024], and MASAC, a discrete-time method of the actor-critic class—for further investigation. The results in Table 1 reconfirm our viewpoint in the *sampling* environment.

---

[5]The repository includes code: `https://github.com/hh11813/continuous-soft-actor-critic`

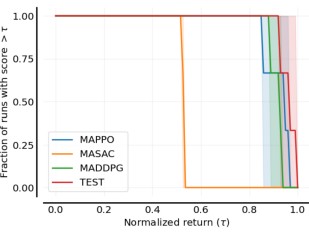
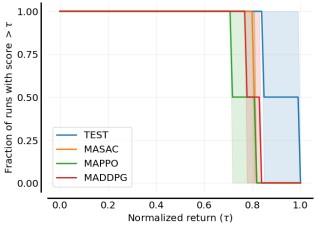

| (a) Navigation. | (b) Sampling. |

Figure 2: Performance profile for two VMAS tasks.

(ii) In contrast, the performance of CMASAC remains stable. Results are shown in Table 2 and 3. Figures 2 present the performances of algorithms MAPPO, MASAC, MADDPG and TEST (a time scaling-free variant of CMASAC, ensuring parameter consistency for cross-algorithm fairness) at $\delta t = 0.01$. The results demonstrate that the martingale approach enhances the algorithm's robustness against small $\delta t$. [6]

(iii) To ensure the comprehensiveness of the experiments and eliminate the impact of extraneous factors on the conclusions, we increased the number of random seeds to five, normalized the rewards, and adjusted the hyperparameters accordingly. The results are summarized in Table 5. The experimental results in Table 5 verify that the temporal robustness of the method proposed in this paper outperforms other methods.

We also conducted a set of experiments in a single-agent environment, which demonstrate the advantages of the off-policy approach. For detailed results, please refer to Appendix C.

## 5 Limitations and future work

Given that the problem formulation is based on stochastic differential equations and constrained by computational resources, we limit our experimental validation to selected benchmarks. The proposed method targets continuous space-time problems and thus may not be suitable for discrete-valued spaces. The theory of multi-agent reinforcement learning requires further exploration. This work serves as an exploratory investigation into continuous-time multi-agent reinforcement learning problems.

## 6 Conclusion

We propose Continuous Soft Actor-Critic, a novel off-policy reinforcement learning algorithm designed for stochastic continuous-time environments, and further generalize it to multi-agent settings. By bridging principles between stochastic optimal control theory and reinforcement learning, we address critical limitations of existing algorithms in continuous-time environments and preliminarily validate our claims. Central to our approach is the enforcement of the martingale property for value functions, coupled with gradient and hyperparameter scaling laws. This results in a $\delta t$-robust parameter update rule. The off-policy nature of the algorithm ensures high sample efficiency, as it enables reuse of historical trajectories. The experimental results on benchmark tasks demonstrate the proposed algorithm's robustness to time discretization.

## Acknowledgments and Disclosure of Funding

This work was supported by the National Key R&D Program of China (NO. 2023YFA1008701) and the Key Project of the National Natural Science Foundation of China (No. 12431017).

---

[6]Further experimental details, scenario-specific analyses, and extended results in different tasks are provided in the Appendix D.

Table 1: Aggregate scores under sampling with different $\delta t$

| $\delta t$ | MASAC(0.1) | MADDPG(0.1) | MASAC(0.01) | MADDPG(0.01) |
|---|---|---|---|---|
| Median | 0.66 [0.62, 0.7] | 0.94 [0.86, 1.0] | 0.43 [0.37, 0.46] | 0.91 [0.79, 1.0] |
| IQM | 0.66 [0.62, 0.7] | 0.94 [0.86, 1.0] | 0.43 [0.37, 0.46] | 0.91 [0.79, 1.0] |
| Mean | 0.66 [0.62, 0.7] | 0.94 [0.86, 1.0] | 0.43 [0.37, 0.46] | 0.91 [0.79, 1.0] |
| Optimality Gap | 0.34 [0.3, 0.38] | 0.06 [0.0, 0.14] | 0.57 [0.54, 0.63] | 0.09 [0.0, 0.21] |

Table 2: Aggregate scores of CMASAC under navigation

| $\delta t$ | 0.1 | 0.01 |
|---|---|---|
| IQM | 0.96 [0.92, 1.0] | 0.93 [0.86, 1.0] |
| Mean | 0.96 [0.92, 1.0] | 0.93 [0.86, 1.0] |
| Median | 0.96 [0.92, 1.0] | 0.93 [0.86, 1.0] |
| Optimality Gap | 0.04 [0.0, 0.08] | 0.07 [0.0, 0.14] |

Table 3: Aggregate scores of CMASAC under sampling

| $\delta t$ | 0.1 | 0.01 |
|---|---|---|
| IQM | 0.84 [0.64, 1.0] | 0.92 [0.83, 1.0] |
| Mean | 0.84 [0.64, 1.0] | 0.92 [0.83, 1.0] |
| Median | 0.84 [0.64, 1.0] | 0.92 [0.83, 1.0] |
| Optimality Gap | 0.16 [0.0, 0.36] | 0.08 [0.0, 0.17] |

Table 4: Aggregate scores under navigation with different $\delta t$

| $\delta t$=0.1 | QMIX | IQL | MAPPO | MASAC |
|---|---|---|---|---|
| Median | 0.92 [0.9, 0.93] | 0.99 [0.98, 1.0] | 0.97 [0.96, 0.98] | 0.89 [0.89, 0.89] |
| IQM | 0.92 [0.9, 0.93] | 0.99 [0.98, 1.0] | 0.97 [0.96, 0.98] | 0.89 [0.89, 0.89] |
| Mean | 0.92 [0.9, 0.93] | 0.99 [0.98, 1.0] | 0.97 [0.96, 0.98] | 0.89 [0.89, 0.89] |
| Optimality Gap | 0.08 [0.07, 0.1] | 0.01 [0.0, 0.02] | 0.03 [0.02, 0.04] | 0.11 [0.11, 0.11] |
| $\delta t$=0.01 | QMIX | IQL | MAPPO | MASAC |
| Median | 0.86 [0.63, 1.0] | 0.88 [0.83, 0.95] | 0.8 [0.73, 0.84] | 0.51 [0.5, 0.52] |
| IQM | 0.86 [0.63, 1.0] | 0.88 [0.83, 0.95] | 0.8 [0.73, 0.84] | 0.51 [0.5, 0.52] |
| Mean | 0.86 [0.63, 1.0] | 0.88 [0.83, 0.95] | 0.8 [0.73, 0.84] | 0.51 [0.5, 0.52] |
| Optimality Gap | 0.14 [0.0, 0.37] | 0.12 [0.05, 0.17] | 0.2 [0.16, 0.27] | 0.49 [0.48, 0.5] |
| $\delta t$ | MADDPG(0.1) | MADDPG(0.01) | | |
| Median | 0.8 [0.76, 0.84] | 0.78 [0.62, 0.91] | | |
| IQM | 0.8 [0.76, 0.84] | 0.78 [0.62, 0.91] | | |
| Mean | 0.8 [0.76, 0.84] | 0.78 [0.62, 0.91] | | |
| Optimality Gap | 0.2 [0.16, 0.24] | 0.22 [0.09, 0.38] | | |

Table 5: Aggregate scores under navigation with $\delta t = 0.01$

| | QMIX | IQL | MAPPO | MASAC |
|---|---|---|---|---|
| Median | 0.75 [0.69, 0.81] | 0.92 [0.89, 0.94] | 0.87 [0.87, 0.89] | 0.49 [0.48, 0.5] |
| IQM | 0.74 [0.67, 0.83] | 0.92 [0.88, 0.95] | 0.87 [0.86, 0.89] | 0.49 [0.47, 0.5] |
| Mean | 0.75 [0.69, 0.81] | 0.92 [0.89, 0.94] | 0.87 [0.87, 0.89] | 0.49 [0.48, 0.5] |
| Optimality Gap | 0.25 [0.19, 0.31] | 0.08 [0.06, 0.11] | 0.13 [0.11, 0.13] | 0.51 [0.5, 0.52] |
| | MADDPG | | | |
| Median | 0.88 [0.79, 0.96] | | | |
| IQM | 0.9 [0.76, 0.98] | | | |
| Mean | 0.88 [0.79, 0.96] | | | |
| Optimality Gap | 0.12 [0.04, 0.21] | | | |

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

# A  Algorithm

Algorithm 2 illustrates the workflow of the Continuous Multi-Agent Soft Actor-Critic.

# B  Proofs

Throughout the proofs, by convention we denote by $A \circ B$ the inner product between $A$ and $B$, by $\|x\|_2$ the Euclidean norm of $x$, $\|A\|_F$ the Frobenius norm of $A$, and $A'$ is transpose of $A$. We denote by $a \vee b$ the larger of a and b, and by $a \wedge b$ the smaller of the two numbers. We denote by $\mathbb{I}$ the indicator function, $\mathbb{I}_{\mathcal{A}}(x) = 1$ when $x \in \mathcal{A}$ and $\mathbb{I}_{\mathcal{A}}(x) = 0$ when $x \notin \mathcal{A}$. For a measurable set $U$, we denote by $\mathcal{P}(U)$ the set of probability distributions over $U$. We assume the following conditions hold to maintain the well-posedness of our problem.

**Assumption 2.** *(i) $b, \bar{\sigma}, r_i, h_i, i = 1, 2$ are all continuous functions in their respective arguments;*

*(ii) $b, \bar{\sigma}$ are uniformly Lipschitz continuous in $x$, i.e., for $\varphi \in \{b, \bar{\sigma}\}$, there exists a constant $C > 0$ such that*

$$|\varphi(t, x, u, v) - \varphi(t, x', u, v)| \leq C |x - x'|, \quad \forall (t, u, v) \in [0, T] \times U \times V, \quad \forall x, x' \in \mathbb{R}^n;$$

*(iii) $b, \bar{\sigma}$ have linear growth in $x$, i.e., for $\varphi \in \{b, \bar{\sigma}\}$, there exists a constant $C > 0$ such that*

$$|\varphi(t, x, u, v)| \leq C(1 + |x|), \quad \forall (t, x, u, v) \in [0, T] \times \mathbb{R}^n \times U \times V;$$

*(iv) $r_i$ and $h_i$ have polynomial growth in $(x, u, v)$ and $x$ respectively, i.e., there exists a constant $C > 0$ and $\mu \geq 1$ such that*

$$|r_i(t, x, u, v)| \leq C \left(1 + |x|^{\mu} + |u|^{\mu} + |v|^{\mu}\right), \quad i = 1, 2,$$
$$|h_i(x)| \leq C \left(1 + |x|^{\mu}\right), \quad \forall (t, x, u, v) \in [0, T] \times \mathbb{R}^n \times U \times V.$$

The following gives the precise definition of admissible policies.

**Definition 1.** *A policy $\pi = \pi(\cdot \mid \cdot, \cdot)$ is called admissible if*

*(i) $\pi(\cdot \mid t, x) \in \mathcal{P}(U), \operatorname{supp} \pi(\cdot \mid t, x) = U$ for every $(t, x) \in [0, T] \times \mathbb{R}^n$, and $\pi(u \mid t, x) : (t, x, u) \in [0, T] \times \mathbb{R}^n \times U \to \mathbb{R}$ is measurable;*

*(ii) $\pi(u \mid t, x)$ is continuous in $(t, x)$ and uniformly Lipschitz continuous in $x$ in the total variation distance, i.e., $\int_U |\pi(u \mid t, x) - \pi(u \mid r, x')| du \to 0$ as $(r, x') \to (t, x)$, and there is a constant $C > 0$ independent of $(t, u)$ such that*

$$\int_U |\pi(u \mid t, x) - \pi(u \mid t, x')| du \leq C |x - x'|, \quad \forall x, x' \in \mathbb{R}^n;$$

*(iii) For any given $\alpha > 0$, the entropy of $\pi$ and its $\alpha$-moment have polynomial growth in $x$, i.e., there are constants $C = C(\alpha) > 0$ and $\mu' = \mu'(\alpha) \geq 1$ such that $\left|\int_U -\log \pi(u \mid t, x)\pi(u \mid t, x)du\right| \leq C \left(1 + |x|^{\mu'}\right)$, and $\int_U |u|^{\alpha}\pi(u \mid t, x)du \leq C \left(1 + |x|^{\mu'}\right), \quad \forall (t, x) \in [0, T] \times \mathbb{R}^n.$*

Under Assumption 2 along with Definition 1, the well-poseness of problems (1)-(2) and (3)-(4) can be guaranteed.

Obviously problem (1)-(2) and (3)-(4) related to stochastic optimal control problem and stochastic game respectively, but the probability space is no longer $(\Omega, \mathcal{F}^W, \mathbb{P}^W)$ but $(\Omega, \mathcal{F}, \mathbb{P})$. [7] We recall the existing results for stochastic control in $(\Omega, \mathcal{F}^W, \mathbb{P}^W)$. Value function $J$ can be characterized by a PDE based on the celebrated Feynman–Kac formula:

$$\begin{cases} \dfrac{\partial J}{\partial t}(t, x) + b(t, x) \circ \dfrac{\partial J}{\partial x}(t, x) + \dfrac{1}{2}\bar{\sigma}^2(t, x) \circ \dfrac{\partial^2 J}{\partial x^2}(t, x) + r(t, x) = 0, \\ J(T, x) = h(x), \end{cases}$$

where $\frac{\partial J}{\partial x} \in \mathbb{R}^n$ is the gradient, and $\frac{\partial^2 J}{\partial x^2} \in \mathbb{R}^{n \times n}$ is the Hessian.

---

[7] Readers may refer to Jia and Zhou [2025] for further discussion on extended probability spaces, which does not affect the methodology of this paper and thus is not elaborated here.

**Algorithm 2** Continuous Multi-Agent Soft Actor-Critic Algorithm

---

**Inputs:** time step $\delta t$, number of epochs $N$, number of mesh grids $K$, number of gradient step $L$, batch size $I$, initial learning rates $\alpha_\theta, \alpha_\phi$, initial $\theta_1, \theta_2, \phi_1, \phi_2, \theta_1^{\text{target}}, \theta_2^{\text{target}}$, discount factor $\beta$, and temperature parameter $\lambda_1, \lambda_2$. $J_1^{\theta_1}(\cdot, \cdot), J_2^{\theta_2}(\cdot, \cdot)$ defining functional form of the value function, $\pi_{\phi_1}^u(\cdot \mid \cdot), \pi_{\phi_2}^v(\cdot \mid \cdot)$ defining functional form of the policy, $\mathcal{D}$ defining buffer of transitions, $\boldsymbol{opt}_J, \boldsymbol{opt}_\pi$ defining optimizer.

**Interactive program:** an environment simulator $(x', r_1, r_2) = \text{Environment}_{\delta t}(t, x, u, v)$ that takes current time-state pair $(t, x)$ and action $u, v$ as inputs and generates state $x'$ at time $t + \delta t$ and the instantaneous reward $r_1, r_2$ at time $t$.

**Learning procedure:**
**for** $j = 1$ **to** $N$ **do**
    Initialize $k = 0$. Observe the initial state $x_0$ and store $x_{t_k} \leftarrow x_0$.
    **while** $k < K$ **do**
        Generate action $v_{t_k} \sim \pi_{\phi_1}^v(\cdot \mid t_k, x_{t_k}), u_{t_k} \sim \pi_{\phi_2}^u(\cdot \mid t_k, x_{t_k})$.
        Apply $v_{t_k}, u_{t_k}$ to the environment simulator $(x, r_1, r_2) = \text{Environment}_{\delta t}(t_k, x_{t_k}, u_{t_k}, v_{t_k})$, and observe the output new state $x$ and reward $r_1, r_2$.
        Store $x_{t_{k+1}} \leftarrow x$ and $r_1^{t_k} \leftarrow r_1, r_2^{t_k} \leftarrow r_2$ in $\mathcal{D}, \mathcal{D} = \mathcal{D} \cup (x_{t_k}, u_{t_k}, v_{t_k}, r_{t_k}, x_{t_{k+1}}, d_{t_{k+1}})$. Update $k \leftarrow k + 1$.
    **end while**
    **for** $l = 1$ **to** $L$ **do**
        Sampled a batch of $I$ transitions $(x_{t_i}, u_{t_i}, v_{t_i}, x_{t_{i+1}}, r_1^{t_i})$ from $\mathcal{D}$ and a batch of $v_{t_i}^{\phi_1}$ from $\pi_{\phi_1}^v$. Compute

$$
\begin{aligned}
\delta M_1 =& J_1^{\theta_1^{\text{target}}}\left(t_{i+1}, x_{t_{i+1}}\right) - J_1^{\theta_1}\left(t_i, x_{t_i}\right) + r_1^{t_i}\delta t - \lambda_1 \ln\left(\pi_{\phi_1}\left(v_{t_i} \mid t_i, x_{t_i}\right)\right)\delta t \\
& - \beta J_1^{\theta_1}\left(t_i, x_{t_i}\right)\delta t,
\end{aligned}
$$

$$
\Delta\theta_1 = \sum_{i=0}^{I-1} \frac{1}{I}\frac{\partial_{\theta_1} J_1^{\theta_1}\left(t_i, x_{t_i}\right)\delta M_1}{\delta t},
$$

$$
\Delta\phi_1 = \sum_{i=0}^{I-1}\frac{1}{I}\frac{\left[\partial_{\phi_1}\ln\pi_{\phi_1}^v\left(v_{t_i}^{\phi_1} \mid t_i, x_{t_i}\right)\right]\delta M_1 - \lambda_1 \partial_{\phi_1}\ln\left(\pi_{\phi_1}^v\left(v_{t_i}^{\phi_1} \mid t_i, x_{t_i}\right)\right)\delta t}{\delta t}.
$$

        Update $\theta_1$ (policy evaluation) with $\boldsymbol{opt}_J$, learning rate $\alpha_\theta \delta t$, and $\Delta\theta_1$.
        Update $\phi_1$ (policy gradient) with $\boldsymbol{opt}_\pi$, learning rate $\alpha_\phi \delta t$ and $\Delta\phi_1$.
        Adjust temperature $\lambda_1$. Update $\theta_1^{\text{target}} \leftarrow \tau\theta_1 + (1-\tau)\theta_1^{\text{target}}$.
        Sampled a batch of $I$ transitions $(x_{t_i}, u_{t_i}, v_{t_i}, x_{t_{i+1}}, r_2^{t_i})$ from $\mathcal{D}$ and a batch of $u_{t_i}^{\phi_2}$ from $\pi_{\phi_2}^u$. Compute

$$
\begin{aligned}
\delta M_2 =& J_2^{\theta_2^{\text{target}}}\left(t_{i+1}, x_{t_{i+1}}\right) - J_2^{\theta_2}\left(t_i, x_{t_i}\right) + r_2^{t_i}\delta t - \lambda_2 \ln\left(\pi_{\phi_2}\left(u_{t_i} \mid t_i, x_{t_i}\right)\right)\delta t \\
& - \beta J_2^{\theta_2}\left(t_i, x_{t_i}\right)\delta t,
\end{aligned}
$$

$$
\Delta\theta_2 = \sum_{i=0}^{I-1} \frac{1}{I}\frac{\partial_{\theta_2} J_2^{\theta_2}(t_i, x_{t_i})\delta M_2}{\delta t},
$$

$$
\Delta\phi_2 = \sum_{i=0}^{I-1}\frac{1}{I}\frac{\left[\partial_{\phi_2}\ln\pi_{\phi_2}^u\left(u_{t_i}^{\phi_2} \mid t_i, x_{t_i}\right)\right]\delta M_2 - \lambda_2 \partial_{\phi_2}\ln\left(\pi_{\phi_2}^u\left(u_{t_i}^{\phi_2} \mid t_i, x_{t_i}\right)\right)\delta t}{\delta t}.
$$

        Update $\theta_2$ with $\boldsymbol{opt}_J$, learning rate $\alpha_\theta \delta t$, and $\Delta\theta_2$.
        Update $\phi_2$ with $\boldsymbol{opt}_\pi$, learning rate $\alpha_\phi \delta t$ and $\Delta\phi_2$.
        Adjust temperature $\lambda_2$. Update $\theta_2^{\text{target}} \leftarrow \tau\theta_2 + (1-\tau)\theta_2^{\text{target}}$.
    **end for**
**end for**

---

## B.1 Proof of Theorem 1

*Proof.* Let $J$ be the value function with policy $\pi \in \mathcal{P}(U)$, applying Itô's lemma to the process $e^{-\beta s}J\left(s, X_s^{\tilde{\pi}}\right)$, we obtain for $0 \le t < s \le T$,

$$e^{-\beta s}J\left(s, X_s^{\tilde{\pi}}; \pi\right) - e^{-\beta t}J(t, x; \pi) + \int_t^s e^{-\beta s'}\left[r\left(s', X_{s'}^{\tilde{\pi}}, u_{s'}^{\tilde{\pi}}\right) - \lambda\ln\pi(u_{s'}^{\tilde{\pi}}|s', X_{s'}^{\tilde{\pi}})\right]ds'$$

$$= \int_t^s e^{-\beta s'}\left[\frac{\partial J}{\partial t}\left(s', X_{s'}^{\tilde{\pi}}; \pi\right) + H\left(s', X_{s'}^{\tilde{\pi}}, u_{s'}^{\tilde{\pi}}, \frac{\partial J}{\partial x}\left(s', X_{s'}^{\tilde{\pi}}; \pi\right), \frac{\partial^2 J}{\partial x^2}\left(s', X_{s'}^{\tilde{\pi}}; \pi\right)\right)\right.$$

$$\left. - \lambda\ln\pi(u_{s'}^{\tilde{\pi}}|s', X_{s'}^{\tilde{\pi}}) - \beta J\left(s', X_{s'}^{\tilde{\pi}}; \pi\right)\right]ds' + \int_t^s e^{-\beta s'}\frac{\partial}{\partial x}J\left(s', X_{s'}^{\tilde{\pi}}; \pi\right) \circ \bar{\sigma}\left(s', X_{s'}^{\tilde{\pi}}, u_{s'}^{\tilde{\pi}}\right)dW_{s'} \tag{20}$$

where for any $(t, x, u) \in [0, T] \times \mathbb{R}^n \times U$,

$$H\left(t, x, u, \frac{\partial J}{\partial x}(t, x; \pi), \frac{\partial^2 J}{\partial x^2}(t, x; \pi)\right)$$

$$= b(t, x, u) \circ \frac{\partial J}{\partial x}(t, x; \pi) + \frac{1}{2}\bar{\sigma}^2(t, x, u) \circ \frac{\partial^2 J}{\partial x^2}(t, x; \pi) + r(t, x, u).$$

Since process (11) with $J$ keeps martingalety, and the term $\int \cdots dW_{s'}$ of the right hand side of (20) is local martingale, we derive that the first term of the right hand side of (20) is also a local martingale. For continuous local martingale with finite variation, by Chapter 1, Exercise 5.21, on Karatzas and Shreve [2014],

$$\int_t^s e^{-\beta s'}\left[\frac{\partial J}{\partial t}\left(s', X_{s'}^{\tilde{\pi}}; \pi\right) + H\left(s', X_{s'}^{\tilde{\pi}}, u_{s'}^{\tilde{\pi}}, \frac{\partial J}{\partial x}\left(s', X_{s'}^{\tilde{\pi}}; \pi\right), \frac{\partial^2 J}{\partial x^2}\left(s', X_{s'}^{\tilde{\pi}}; \pi\right)\right)\right.$$

$$\left. - \lambda\ln\pi(u_{s'}^{\tilde{\pi}}|s', X_{s'}^{\tilde{\pi}}) - \beta J\left(s', X_{s'}^{\tilde{\pi}}; \pi\right)\right]ds' = 0, \quad \forall s \in [t, T], \tag{21}$$

$\mathbb{P}$-almost surely holds.

Denote

$$f(t, x, u) = \frac{\partial J}{\partial t}(t, x) + H\left(t, x, u, \frac{\partial J}{\partial x}(t, x), \frac{\partial^2 J}{\partial x^2}(t, x)\right) - \lambda\ln\pi(u|t, x) - \beta J(t, x).$$

We shall demonstrate that

$$f(t, x, u) = 0, \quad \forall(t, x, u) \in [0, T] \times \mathbb{R}^n \times U. \tag{22}$$

Since $f$ is a continuous function, if (22) is not true, there exists $(t^*, x^*, u^*)$ and $\epsilon > 0$ such that $|f(t^*, x^*, u^*)| > \epsilon$. Without loss of generality, we assume $f(t^*, x^*, u^*) > \epsilon$. Then exists $\delta > 0$ such that $f(r, x', u') > \epsilon/2$ for all $(r, x', u')$ with $|r - t^*| \vee |x' - x^*| \vee |u' - u^*| < \delta$. Consider the state process $X^{\tilde{\pi}}$, starting from $(t^*, x^*, u^*)$, namely, $\{X_s^{\tilde{\pi}}, t^* \le s \le T\}$ follows (1) with $X_{t^*}^{\tilde{\pi}} = x^*$ and $u_{t^*}^{\tilde{\pi}} = u^*$. Define

$$\tau = \inf\left\{r \ge t^* : |r - t^*| > \delta \text{ or } \left|X_r^{\tilde{\pi}} - x^*\right| > \delta\right\} = \inf\left\{r \ge t^* : \left|X_r^{\tilde{\pi}} - x^*\right| > \delta\right\} \wedge (t^* + \delta).$$

The continuity of $X^{\tilde{\pi}}$ implies that $\tau > t^*$, $\mathbb{P}$-almost surely.

(21) means that there exists $\Omega_0 \in \mathcal{F}$ with $\mathbb{P}(\Omega_0) = 0$ such that for all $\omega \in \Omega\backslash\Omega_0$ and $s \in [t^*, T]$, $\int_{t^*}^s e^{-\beta s'}f\left(s', X_{s'}^{\tilde{\pi}}(\omega), u_{s'}^{\tilde{\pi}}(\omega)\right)ds' = 0$. It follows from Lebesgue's differentiation theorem that for any $\omega \in \Omega\backslash\Omega_0$,

$$f\left(s, X_s^{\tilde{\pi}}(\omega), u_s^{\tilde{\pi}}(\omega)\right) = 0, \text{ a.e. } s \in [t^*, \tau(\omega)].$$

Consider the set $Z(\omega) = \left\{s \in [t^*, \tau(\omega)] : u_s^{\tilde{\pi}}(\omega) \in \mathcal{B}_\delta(u^*)\right\} \subset [t^*, \tau(\omega)]$, where $\mathcal{B}_\delta(u^*) = \{u' \in U : |u' - u^*| < \delta\}$ is the neighborhood of $u^*$. Because $f\left(s, X_s^{\tilde{\pi}}(\omega), u_s^{\tilde{\pi}}(\omega)\right) > \frac{\epsilon}{2}$ when $s \in Z(\omega)$, we conclude that $Z(\omega)$ has Lebesgue measure zero for any $\omega \in \Omega\backslash\Omega_0$. That is,

$$\int_{[t^*, T]}\mathbb{I}_{\{s \le \tau(\omega)\}}\mathbb{I}_{\{u_s^{\tilde{\pi}}(\omega)\in\mathcal{B}_\delta(u^*)\}}ds = 0.$$

Integrating $\omega$ with respect to $\mathbb{P}$ and applying Fubini's theorem, we obtain

$$
\begin{aligned}
0 &= \int_\Omega \int_{[t^*,T]} \mathbb{I}_{\{s \le \tau(\omega)\}} \mathbb{I}_{\{u_s^{\tilde{\pi}}(\omega) \in \mathcal{B}_\delta(u^*)\}} ds \mathbb{P}(d\omega) = \int_{[t^*,T]} \mathbb{E}\left[\mathbb{I}_{\{s \le \tau\}} \mathbb{I}_{\{u_s^{\tilde{\pi}} \in \mathcal{B}_\delta(u^*)\}}\right] ds \\
&= \int_{t^*}^T \mathbb{E}\left[\mathbb{I}_{\{s \le \tau\}} \mathbb{P}\left(u_s^{\tilde{\pi}} \in \mathcal{B}_\delta(u^*) \mid \mathcal{F}_s\right)\right] ds = \int_{t^*}^T \mathbb{E}\left[\mathbb{I}_{\{s \le \tau\}} \int_{\mathcal{B}_\delta(u^*)} \tilde{\pi}\left(u \mid s, X_s^{\tilde{\pi}}\right) du\right] ds \\
&\ge \min_{|x'-x^*|<\delta, |r-t^*|<\delta} \left\{\int_{\mathcal{B}_\delta(u^*)} \tilde{\pi}\left(u \mid r, x'\right) du\right\} \int_{t^*}^T \mathbb{E}\left[\mathbb{I}_{\{s \le \tau\}}\right] ds \\
&= \min_{|x'-x^*|<\delta, |r-t^*|<\delta} \left\{\int_{\mathcal{B}_\delta(u^*)} \tilde{\pi}\left(u \mid r, x'\right) du\right\} \mathbb{E}\left[(\tau \wedge T) - t^*\right] \ge 0.
\end{aligned}
$$

Since $\tau > t^*$, $\mathbb{P}$-almost surely, the above implies

$$
\min_{|x'-x^*|<\delta, |r-t^*|<\delta} \left\{\int_{\mathcal{B}_\delta(u^*)} \tilde{\pi}\left(u \mid r, x'\right) du\right\} = 0.
$$

However, this contradicts Definition 1 about an admissible policy $\tilde{\pi}$. Indeed, Definition 1-(i) stipulates $\operatorname{supp} \tilde{\pi}(\cdot \mid t, x) = U$ for any $(t, x)$, hence $\int_{\mathcal{B}_\delta(u^*)} \tilde{\pi}(u \mid t, x) du > 0$. Then the continuity in Definition 1-(ii) yields

$$
\min_{|x'-x^*|<\delta, |r-t^*|<\delta} \left\{\int_{\mathcal{B}_\delta(u^*)} \tilde{\pi}\left(u \mid r, x'\right) du\right\} > 0,
$$

it is a contradiction. Hence we conclude $f(t, x, u) = 0$ for every $(t, x, u) \in [0, T] \times \mathbb{R}^n \times U$. Then we have

$$
\begin{aligned}
\int_U &\left[\frac{\partial J}{\partial t}(t, x) + b(t, x, u) \circ \frac{\partial J}{\partial x}(t, x) + \frac{1}{2}\bar{\sigma}^2(t, x, u) \circ \frac{\partial^2 J}{\partial x^2}(t, x)\right. \\
&\left. + r(t, x, u) - \lambda \ln \pi(u \mid t, x) - \beta J(t, x)\right] \pi(u \mid t, x) du = 0.
\end{aligned}
$$

Assume that the function $J^\theta$ satisfies the assumptions of Theorem 1. Combining the above results with the hypotheses of Theorem 1, we conclude that $J^\theta$ satisfies:

$$
\begin{cases}
\dfrac{\partial J^\theta}{\partial t}(t, x) + \displaystyle\int_U \left[b(t, x, u) \circ \dfrac{\partial J^\theta}{\partial x}(t, x) + \dfrac{1}{2}\bar{\sigma}^2(t, x, u) \circ \dfrac{\partial^2 J^\theta}{\partial x^2}(t, x)\right. \\
\left. + r(t, x, u) - \lambda \ln \pi(u \mid t, x)\right] \pi(u \mid t, x) du - \beta J^\theta(t, x) = 0, \\
J^\theta(T, x) = h(x).
\end{cases}
\tag{23}
$$

Then $J^\theta$ is the unique viscosity solution among polynomially growing functions of (23). $J^\theta$ is the value function associated with policy $\pi$. $\qquad\square$

## B.2 Proof of Theorem 2

Consider system:

$$
\begin{cases}
d\tilde{X}^\pi(t) = \tilde{b}\left(t, \tilde{X}^\pi(t), \bar{\pi}_t, v_t\right) dt + \tilde{\sigma}\left(t, \tilde{X}^\pi(t), \bar{\pi}_t, v_t\right) dW(t), \\
\tilde{X}^\pi(0) = x_0 \in \mathbb{R}^n, \quad t \ge 0
\end{cases}
\tag{24}
$$

where

$$
\tilde{b}(s, x, \pi(\cdot), v) = \int_U b(s, x, u, v)\pi(u) du, \quad \tilde{\sigma}(s, x, \pi(\cdot), v) = \sqrt{\int_U \bar{\sigma}^2(s, x, u, v)\pi(u) du}.
$$

Value function with policies $(\bar{\pi}^u, \pi^v)$ admits the following representation:

$$
\begin{aligned}
J(t, x; \pi) = \mathbb{E}_{\mathbb{P}}\left[\int_t^T e^{-\beta(s-t)} \left\{\int_U \left[r(t, \tilde{X}_t^\pi, u, v)\bar{\pi}^u(u|t, \tilde{X}_t^\pi) - \lambda \ln \pi^v(v|t, \tilde{X}_t^\pi)\right]\right.\right. \\
\left.\left. \pi^v(v|t, \tilde{X}_t^\pi) du\right\} ds + e^{-\beta(T-t)} h(\tilde{X}_T^\pi) \big| \tilde{X}_t = x\right],
\end{aligned}
\tag{25}
$$

The problem (24)-(25) is equavilent to (3)-(4) and the solution of the SDE (3) $X^\pi$ with policies $(\bar{\pi}^u, \pi^v)$ shares the same distribution as $\tilde{X}^\pi$ of (24).

*Proof.* Considering the equivalent formulation (24)-(25), by applying the same method for process $e^{-\beta s} J_1\left(s, \tilde{X}_s^{\tilde{\pi}}\right)$ with policy $\pi = (\bar{\pi}^u, \pi^v)$ as used in the proof of Theorem 1, we can get for $0 \leq t < s \leq T$:

$$
e^{-\beta s} J_1\left(s, \tilde{X}_s^{\tilde{\pi}}; \pi\right) - e^{-\beta t} J_1(t, x; \pi)
$$
$$
+ \int_t^s e^{-\beta s'} \left( \int_U r_1(s', \tilde{X}_{s'}^{\tilde{\pi}}, u_{s'}^{\bar{\pi}}, v_{s'}^{\tilde{\pi}}) \bar{\pi}^u(u^{\bar{\pi}}|s', \tilde{X}_{s'}^{\tilde{\pi}}) du - \lambda_1 \ln \pi^v(v_{s'}^{\tilde{\pi}}|s', \tilde{X}_{s'}^{\tilde{\pi}}) \right) ds'
$$
$$
= \int_t^s e^{-\beta s'} \left[ \frac{\partial J_1}{\partial t}\left(s', \tilde{X}_{s'}^{\tilde{\pi}}; \pi\right) + \int_U H_1\left(s', \tilde{X}_{s'}^{\tilde{\pi}}, u_{s'}^{\bar{\pi}}, v_{s'}^{\tilde{\pi}}, \frac{\partial J_1}{\partial x}\left(s', \tilde{X}_{s'}^{\tilde{\pi}}; \pi\right), \frac{\partial^2 J_1}{\partial x^2}\left(s', \tilde{X}_{s'}^{\tilde{\pi}}; \pi\right) \right) \right.
$$
$$
\left. \bar{\pi}^u(u^{\bar{\pi}}|s', \tilde{X}_{s'}^{\tilde{\pi}}) du - \lambda_1 \ln \pi^v(v_{s'}^{\tilde{\pi}}|s', \tilde{X}_{s'}^{\tilde{\pi}}) - \beta J_1\left(s', \tilde{X}_{s'}^{\tilde{\pi}}; \pi\right) \right] ds'
$$
$$
+ \int_t^s e^{-\beta s'} \frac{\partial}{\partial x} J_1\left(s', \tilde{X}_{s'}^{\tilde{\pi}}; \pi\right) \circ \bar{\sigma}\left(s', \tilde{X}_{s'}^{\tilde{\pi}}, u_{s'}^{\bar{\pi}}, v_{s'}^{\tilde{\pi}}\right) dW_{s'}
$$
(26)

where

$$
H_1\left(s', \tilde{X}_{s'}^{\tilde{\pi}}, u^{\bar{\pi}}, v^{\tilde{\pi}}, \frac{\partial J_1}{\partial x}\left(s', \tilde{X}_{s'}^{\tilde{\pi}}; \pi\right), \frac{\partial^2 J_1}{\partial x^2}\left(s', \tilde{X}_{s'}^{\tilde{\pi}}; \pi\right) \right)
$$
$$
= b(s', \tilde{X}_{s'}^{\tilde{\pi}}, u^{\bar{\pi}}, v^{\tilde{\pi}}) \circ \frac{\partial J_1}{\partial x}\left(s', \tilde{X}_{s'}^{\tilde{\pi}}; \pi\right) + \frac{1}{2} \bar{\sigma}^2(s', \tilde{X}_{s'}^{\tilde{\pi}}, u^{\bar{\pi}}, v^{\tilde{\pi}}) \circ \frac{\partial^2 J_1}{\partial x^2}\left(s', \tilde{X}_{s'}^{\tilde{\pi}}; \pi\right)
$$
$$
+ r_1(s', \tilde{X}_{s'}^{\tilde{\pi}}, u^{\bar{\pi}}, v^{\tilde{\pi}}).
$$

Following the methodology of proving Theorem 1, we extend the argument to the multi-agent case. Since process (14) with $J_1$ keeps martingalety, and the term $\int \cdots dW_{s'}$ of the right hand side of (26) is local martingale, we derive that the first term of the right hand side of (26) is also a local martingale. Then

$$
\int_t^s e^{-\beta s'} \left[ \frac{\partial J_1}{\partial t}\left(s', \tilde{X}_{s'}^{\tilde{\pi}}; \pi\right) + \int_U H_1\left(s', \tilde{X}_{s'}^{\tilde{\pi}}, u_{s'}^{\bar{\pi}}, v_{s'}^{\tilde{\pi}}, \frac{\partial J_1}{\partial x}\left(s', \tilde{X}_{s'}^{\tilde{\pi}}; \pi\right), \frac{\partial^2 J_1}{\partial x^2}\left(s', \tilde{X}_{s'}^{\tilde{\pi}}; \pi\right) \right) \right.
$$
$$
\left. \bar{\pi}^u(u^{\bar{\pi}}|s', \tilde{X}_{s'}^{\tilde{\pi}}) du - \lambda_1 \ln \pi^v(v_{s'}^{\tilde{\pi}}|s', \tilde{X}_{s'}^{\tilde{\pi}}) - \beta J_1\left(s', \tilde{X}_{s'}^{\tilde{\pi}}; \pi\right) \right] ds' = 0, \quad \forall s \in [t, T],
$$
(27)

$\mathbb{P}$-almost surely holds.

Denote

$$
f(t, x, v) = \frac{\partial J_1}{\partial t}(t, x) + \int_U H_1\left(t, x, u, v, \frac{\partial J_1}{\partial x}(t, x), \frac{\partial^2 J_1}{\partial x^2}(t, x) \right) \bar{\pi}^u(u \mid t, x) du
$$
$$
- \lambda_1 \ln \pi^v(v|t, x) - \beta J_1(t, x).
$$

We shall demonstrate that

$$
f(t, x, v) = 0, \quad \forall (t, x, v) \in [0, T] \times \mathbb{R}^n \times V.
$$
(28)

Since $f$ is a continuous function, if (28) is not true, there exists $(t^*, x^*, v^*)$ and $\epsilon > 0$ such that $|f(t^*, x^*, v^*)| > \epsilon$. Without loss of generality, we assume $f(t^*, x^*, v^*) > \epsilon$. Then exists $\delta > 0$ such that $f(r, x', v') > \epsilon/2$ for all $(r, x', v')$ with $|r - t^*| \vee |x' - x^*| \vee |v' - v^*| < \delta$. Consider the state process $X^{\tilde{\pi}}$, starting from $(t^*, x^*, v^*)$, namely, $\{X_s^{\tilde{\pi}}, t^* \leq s \leq T\}$ follows (24) with $X_{t^*}^{\tilde{\pi}} = x^*$ and $v_{t^*}^{\tilde{\pi}} = v^*$. Define

$$
\tau = \inf\left\{r \geq t^* : |r - t^*| > \delta \text{ or } \left|X_r^{\tilde{\pi}} - x^*\right| > \delta\right\} = \inf\left\{r \geq t^* : \left|X_r^{\tilde{\pi}} - x^*\right| > \delta\right\} \wedge (t^* + \delta).
$$

The continuity of $X^{\tilde{\pi}}$ implies that $\tau > t^*$, $\mathbb{P}$-almost surely.

(27) means that there exists $\Omega_0 \in \mathcal{F}$ with $\mathbb{P}(\Omega_0) = 0$ such that for all $\omega \in \Omega \backslash \Omega_0$, for all $s \in [t^*, T]$, $\int_{t^*}^s e^{-\beta s'} f\left(s', X_{s'}^{\tilde{\pi}}(\omega), v_{s'}^{\tilde{\pi}}(\omega)\right) ds' = 0$. It follows from Lebesgue's differentiation theorem that for any $\omega \in \Omega \backslash \Omega_0$,

$$
f\left(s, X_s^{\tilde{\pi}}(\omega), v_s^{\tilde{\pi}}(\omega)\right) = 0, \text{ a.e. } s \in [t^*, \tau(\omega)].
$$

Consider the set $Z(\omega) = \left\{ s \in [t^*, \tau(\omega)] : v_s^{\tilde{\pi}}(\omega) \in \mathcal{B}_\delta(v^*) \right\} \subset [t^*, \tau(\omega)]$, where $\mathcal{B}_\delta(v^*) = \{ v' \in V : |v' - v^*| < \delta \}$ is the neighborhood of $v^*$. Because $f\left(s, X_s^{\tilde{\pi}}(\omega), v_s^{\tilde{\pi}}(\omega)\right) > \frac{\epsilon}{2}$ when $s \in Z(\omega)$, we conclude that $Z(\omega)$ has Lebesgue measure zero for any $\omega \in \Omega \backslash \Omega_0$. That is,

$$\int_{[t^*, T]} \mathbb{I}_{\{s \leq \tau(\omega)\}} \mathbb{I}_{\{v_s^{\tilde{\pi}}(\omega) \in \mathcal{B}_\delta(v^*)\}} ds = 0.$$

Integrating $\omega$ with respect to $\mathbb{P}$ and applying Fubini's theorem, we obtain

$$0 = \int_\Omega \int_{[t^*, T]} \mathbb{I}_{\{s \leq \tau(\omega)\}} \mathbb{I}_{\{v_s^{\tilde{\pi}}(\omega) \in \mathcal{B}_\delta(v^*)\}} ds \mathbb{P}(d\omega) = \int_{[t^*, T]} \mathbb{E}\left[\mathbb{I}_{\{s \leq \tau\}} \mathbb{I}_{\{v_s^{\tilde{\pi}} \in \mathcal{B}_\delta(v^*)\}}\right] ds$$

$$= \int_{t^*}^T \mathbb{E}\left[\mathbb{I}_{\{s \leq \tau\}} \mathbb{P}\left(v_s^{\tilde{\pi}} \in \mathcal{B}_\delta(v^*) \mid \mathcal{F}_s\right)\right] ds = \int_{t^*}^T \mathbb{E}\left[\mathbb{I}_{\{s \leq \tau\}} \int_{\mathcal{B}_\delta(v^*)} \tilde{\pi}\left(v \mid s, X_s^{\tilde{\pi}}\right) dv\right] ds$$

$$\geq \min_{|x'-x^*|<\delta, |r-t^*|<\delta} \left\{ \int_{\mathcal{B}_\delta(v^*)} \tilde{\pi}\left(v \mid r, x'\right) dv \right\} \int_{t^*}^T \mathbb{E}\left[\mathbb{I}_{\{s \leq \tau\}}\right] ds$$

$$= \min_{|x'-x^*|<\delta, |r-t^*|<\delta} \left\{ \int_{\mathcal{B}_\delta(v^*)} \tilde{\pi}\left(v \mid r, x'\right) dv \right\} \mathbb{E}\left[(\tau \wedge T) - t^*\right] \geq 0.$$

Since $\tau > t^*$, $\mathbb{P}$-almost surely, the above implies

$$\min_{|x'-x^*|<\delta, |r-t^*|<\delta} \left\{ \int_{\mathcal{B}_\delta(v^*)} \tilde{\pi}\left(v \mid r, x'\right) dv \right\} = 0.$$

However, this contradicts Definition 1 about an admissible policy $\tilde{\pi}$. Indeed, Definition 1-(i) stipulates supp $\tilde{\pi}(\cdot \mid t, x) = V$ for any $(t, x)$, hence $\int_{\mathcal{B}_\delta(v^*)} \tilde{\pi}(v \mid t, x) dv > 0$. Then the continuity in Definition 1-(ii) yields

$$\min_{|x'-x^*|<\delta, |r-t^*|<\delta} \left\{ \int_{\mathcal{B}_\delta(v^*)} \tilde{\pi}\left(v \mid r, x'\right) dv \right\} > 0,$$

it is a contradiction. Hence we conclude $f(t, x, v) = 0$ for every $(t, x, v) \in [0, T] \times \mathbb{R}^n \times V$. Then we have

$$\int_V \left\{ \frac{\partial J_1}{\partial t}(t, x) + \int_U \left[ b(t, x, u, v) \circ \frac{\partial J_1}{\partial x}(t, x) + \frac{1}{2}\bar{\sigma}^2(t, x, u, v) \circ \frac{\partial^2 J_1}{\partial x^2}(t, x) \right. \right.$$
$$\left. \left. + r(t, x, u, v) \right] \bar{\pi}^u(u \mid t, x) du - \lambda_1 \ln \pi^v(v \mid t, x) - \beta J_1(t, x) \right\} \pi^v(v \mid t, x) dv = 0.$$

Assume that the function $J_1^\theta$ satisfies the assumptions of Theorem 2. Combining the above results with the hypotheses of Theorem 2, we conclude that $J_1^\theta$ satisfies:

$$\begin{cases} \frac{\partial J_1^\theta}{\partial t}(t, x) + \int_V \left[ \int_U H_1\left(t, x, u, v, \frac{\partial J_1^\theta}{\partial x}(t, x), \frac{\partial^2 J_1^\theta}{\partial x^2}(t, x)\right) \bar{\pi}^u(u \mid t, x) du \right. \\ \left. -\lambda_1 \ln \pi^v(v|t, x) \right] \pi^v(v \mid t, x) dv - \beta J_1^\theta(t, x) = 0, \\ J_1^\theta(T, x) = h_1(x). \end{cases} \quad (29)$$

Then $J_1^\theta$ is the unique viscosity solution among polynomially growing functions of (29). $J_1^\theta$ is the value function associated with policy $\pi$. The proof for $J_2$ follows a procedure analogous to that of $J_1$. $\qquad \square$

## B.3 Proof of Theorem 3

*Proof.* For value function $J$ with policy $\pi$, we have

$$\int_U \left(\mathcal{L}^u J(t, x) + r(t, x, u) - \lambda \ln \pi(u) - \beta J(t, x)\right) \pi(u) du = 0 \quad (30)$$

where

$$\mathcal{L}^u J(t, x) := \frac{\partial J(t, x)}{\partial t} + b(t, x, u) \circ \frac{\partial J(t, x)}{\partial x} + \frac{1}{2}\bar{\sigma}^2(t, x, u) \circ \frac{\partial^2 J(t, x)}{\partial x^2}.$$

Let $\pi_\phi$ be the parametric family of policies with the parameter $\phi \in \Phi$. We aim to compute the policy gradient $g(t, x; \phi) := \frac{\partial J(t,x;\pi_\phi)}{\partial \phi} \in \mathbb{R}^{L_\phi}$ at the current time–state pair $(t, x)$.

We take the derivative in $\phi$ on both sides of (30) and we have

$$\begin{cases} \int_U \left\{ \left[ \mathcal{L}^u g(t, x; \phi) - \lambda \frac{\partial}{\partial \phi} \ln \pi^\phi(u \mid t, x)) - \beta g(t, x; \phi) \right] \pi^\phi(u) \right. \\ \left. + \left( \mathcal{L}^u J(t, x; \phi) + r(t, x, u) - \lambda \ln \pi^\phi(u) - \beta J(t, x; \phi) \right) \frac{\partial \pi^\phi}{\partial \phi}(u) \right\} du = 0, \\ g(T, x; \phi) = 0. \end{cases} \tag{31}$$

Define

$$\check{r}(t, x, u; \phi)$$

$$= \left[ \mathcal{L}^u J(t, x; \phi) + r(t, x, u) - \lambda \ln \pi^\phi(u) - \beta J(t, x; \phi) \right] \frac{\frac{\partial \pi^\phi(u)}{\partial \phi}}{\pi^\phi(u)} - \lambda \frac{\partial}{\partial \phi} \ln \pi^\phi(u \mid t, x)$$

$$= \left[ \mathcal{L}^u J(t, x; \phi) + r(t, x, u) - \lambda \ln \pi^\phi(u) - \beta J(t, x; \phi) \right] \frac{\partial}{\partial \phi} \ln \pi^\phi(u) - \lambda \frac{\partial}{\partial \phi} \ln \pi^\phi(u \mid t, x).$$

Then (31) can be written as

$$\begin{cases} \int_U \left[ \mathcal{L}^u g(t, x; \phi) + \check{r}(t, x, u; \phi) - \beta g(t, x; \phi) \right] \pi^\phi(u) du = 0, \\ g(T, x; \phi) = 0. \end{cases} \tag{32}$$

The Feynman-Kac formula represents $g(t, x; \phi)$ as

$$g(t, x; \phi) = \mathbb{E}_{\mathbb{P}} \left[ \int_t^T e^{-\beta s} \check{r}(s, X_s, u_s; \phi) \, ds \mid X_t^\pi = x \right]. \tag{33}$$

Apply Itô lemma to $J(s, X_s^\pi; \phi)$, we obtain

$$J\left(t + \delta t, X_{t+\delta t}^\pi; \phi\right) - J\left(t, X_t^\pi; \phi\right) = \int_t^{t+\delta t} \mathcal{L}^u J\left(s, X_s^\pi; \phi\right) ds + \frac{\partial J}{\partial x}\left(s, X_s^\pi; \phi\right)' \circ \bar{\sigma}(s, X_s^\pi, u^\pi) dW_s.$$

Therefore,

$$\check{r}\left(s, X_s^\pi, u_s; \phi\right) ds$$

$$= \left[ \mathcal{L}^u J\left(s, X_s^\pi; \phi\right) + r\left(s, X_s^\pi, u_s\right) - \lambda \ln \pi^\phi(u_s \mid s, X_s^\pi) - \beta J\left(s, X_s^\pi; \phi\right) \right] \times \frac{\partial}{\partial \phi} \ln \pi^\phi\left(u_s \mid s, X_s^\pi\right) ds$$

$$- \lambda \frac{\partial}{\partial \phi} \ln \pi^\phi(u_s \mid s, X_s^\pi) ds$$

$$\approx \frac{\partial}{\partial \phi} \ln \pi^\phi\left(u_s \mid s, X_s^\pi\right) \left\{ dJ\left(s, X_s^\pi; \phi\right) + \left[ r\left(s, X_s^\pi, u_s\right) - \lambda \ln \pi^\phi(u_s \mid s, X_s^\pi) - \beta J\left(s, X_s^\pi; \phi\right) \right] ds \right.$$

$$\left. - \frac{\partial J}{\partial x}\left(s, X_s^\pi; \phi\right)' \circ \bar{\sigma}(s, X_s^\pi, u^\pi) dW_s \right\} - \lambda \frac{\partial}{\partial \phi} \ln \pi^\phi(u_s \mid s, X_s^\pi) ds.$$

$$\tag{34}$$

By substituting equation (34) into equation (33), the theorem is proven. $\qquad\square$

### B.4   Proof of Theorem 4

*Proof.* From the Feymann-Kac formula, we have $J_1(t, x; \bar{\pi}^u, \pi^v)$ with policy $\pi = (\bar{\pi}^u, \pi^v)$ satisfies

$$\int_V \left[ \int_U \left( \mathcal{L}^{u,v} J_1(t, x; \bar{\pi}^u, \pi^v) + r_1(t, x, u, v) \right) \bar{\pi}^u(u) du - \lambda_1 \ln \pi^v(v) \right. \\ \left. - \beta J_1(t, x; \bar{\pi}^u, \pi^v) \right] \pi^v(v) dv = 0 \tag{35}$$

where

$$\mathcal{L}^{u,v} J_1(t, x) := \frac{\partial J_1(t, x)}{\partial t} + b(t, x, u, v) \circ \frac{\partial J_1(t, x)}{\partial x} + \frac{1}{2} \bar{\sigma}^2(t, x, u, v) \circ \frac{\partial^2 J_1(t, x)}{\partial x^2}.$$

Let $(\pi_{\phi_2}^u, \pi_{\phi_1}^v)$ be the parametric family of policies with the parameter $\phi_1 \in \Phi_1 \subset \mathbb{R}^{L_{\phi_1}}$, $\phi_2 \in \Phi_2 \subset$ $\mathbb{R}^{L_{\phi_2}}$. We aim to compute the policy gradient $g_1(t, x; \bar{\phi}_2, \phi_1) := \frac{\partial J_1(t, x; \bar{\pi}_{\phi_2}^u, \pi_{\phi_1}^v)}{\partial \phi_1}$, $g_2(t, x; \phi_2, \bar{\phi}_1) :=$ $\frac{\partial J_2(t, x; \pi_{\phi_2}^u, \bar{\pi}_{\phi_1}^v)}{\partial \phi_2}$.

Taking the derivative in $\phi_1$ on both sides of the HJB equation (35), we obtain:

$$
\begin{cases}
\int_V \left\{ \left[ \int_U \mathcal{L}^{u,v} g_1(t, x; \bar{\phi}_2, \phi_1) \bar{\pi}_{\phi_2}^u(u) du - \lambda_1 \frac{\partial \ln \pi_{\phi_1}^v(v)}{\partial \phi_1} - \beta g_1(t, x; \bar{\phi}_2, \phi_1) \right] \pi_{\phi_1}^v(v) \right. \\
+ \left[ \int_U \left( \mathcal{L}^{u,v} J_1(t, x; \bar{\phi}_2, \phi_1) + r_1(t, x, u, v) \right) \bar{\pi}_{\phi_2}^u(u) du - \lambda_1 \ln \pi_{\phi_1}^v(v) \right. \\
\left. -\beta J_1(t, x; \bar{\phi}_2, \phi_1) \right] \frac{\partial \pi_{\phi_1}^v(v)}{\partial \phi_1} \Big\} dv = 0, \\
g_1(T, x; \bar{\phi}_2, \phi_1) = 0.
\end{cases}
\tag{36}
$$

Define

$$
\check{r}(t, x, u, v; \bar{\phi}_2, \phi_1)
$$

$$
= \left[ \mathcal{L}^{u,v} J_1(t, x; \bar{\phi}_2, \phi_1) + r_1(t, x, u, v) - \lambda_1 \ln \pi_{\phi_1}^v(v) - \beta J_1(t, x; \bar{\phi}_2, \phi_1) \right] \frac{\frac{\partial \pi_{\phi_1}^v(v)}{\partial \phi_1}}{\pi_{\phi_1}^v(v)}
$$

$$
- \lambda_1 \frac{\partial}{\partial \phi_1} \ln \pi_{\phi_1}^v(v)
$$

$$
= \left[ \mathcal{L}^{u,v} J_1(t, x; \bar{\phi}_2, \phi_1) + r_1(t, x, u, v) - \lambda_1 \ln \pi_{\phi_1}^v(v) - \beta J_1(t, x; \bar{\phi}_2, \phi_1) \right] \frac{\partial}{\partial \phi_1} \ln \pi_{\phi_1}^v(v)
$$

$$
- \lambda_1 \frac{\partial}{\partial \phi_1} \ln \pi_{\phi_1}^v(v).
$$

Then we have

$$
\begin{cases}
\int_V \int_U \left[ \mathcal{L}^{u,v} g_1(t, x; \bar{\phi}_2, \phi_1) + \check{r}(t, x, u, v; \bar{\phi}_2, \phi_1) - \beta g_1(t, x; \bar{\phi}_2, \phi_1) \right] \bar{\pi}_{\phi_2}^u(u) du \pi_{\phi_1}^v(v) dv = 0, \\
g_1(T, x; \bar{\phi}_2, \phi_1) = 0.
\end{cases}
$$

Then $g_1$ is represented by

$$
g_1(t, x; \bar{\phi}_2, \phi_1) = \mathbb{E}_{\mathbb{P}} \left[ \int_t^T e^{-\beta s} \check{r}\left( s, X_s, u_s, v_s; \bar{\phi}_2, \phi_1 \right) ds \mid X_t^\pi = x \right].
\tag{37}
$$

Apply Itô lemma to $J_1(t, X_s^\pi; \bar{\phi}_2, \phi_1)$ on $[t, t + \delta t]$,

$$
J_1\left( t + \delta t, X_{t+\delta t}^\pi; \bar{\phi}_2, \phi_1 \right) - J_1\left( t, X_t^\pi; \bar{\phi}_2, \phi_1 \right) = \int_t^{t+\delta t} \mathcal{L}^{u,v} J_1\left( s, X_s^\pi; \bar{\phi}_2, \phi_1 \right) ds
$$

$$
+ \frac{\partial J_1}{\partial x}\left( s, X_s^\pi; \bar{\phi}_2, \phi_1 \right)' \circ \bar{\sigma}(t, X_s^\pi, u^\pi, v^\pi) dW_s.
\tag{38}
$$

Therefore,

$$
\check{r}(s, X_s^\pi, u_s, v_s; \bar{\phi}_2, \phi_1) ds
$$

$$
= \left[ \mathcal{L}^{u,v} J_1\left( s, X_s^\pi; \bar{\phi}_2, \phi_1 \right) + r_1(s, X_s^\pi, u_s, v_s) - \lambda_1 \ln \pi_{\phi_1}^v(v) - \beta J_1(s, X_s^\pi; \bar{\phi}_2, \phi_1) \right] \cdot
$$

$$
\frac{\partial}{\partial \phi_1} \ln \pi_{\phi_1}^v(v) ds - \lambda_1 \frac{\partial}{\partial \phi_1} \ln \pi_{\phi_1}^v(v) ds
$$

$$
\approx \frac{\partial}{\partial \phi_1} \ln \pi_{\phi_1}^v(v) \left\{ dJ_1\left( s, X_s^\pi; \bar{\phi}_2, \phi_1 \right) + \left[ r(s, X_s^\pi, u_s, v_s) - \lambda_1 \ln \pi_{\phi_1}^v(v) - \beta J_1(s, X_s^\pi; \bar{\phi}_2, \phi_1) \right] ds \right.
$$

$$
\left. - \frac{\partial J}{\partial x}\left( s, X_s^\pi; \bar{\phi}_2, \phi_1 \right)' \circ \bar{\sigma}(s, X_s^\pi, u^\pi, v^\pi) dW_s \right\} - \lambda_1 \frac{\partial}{\partial \phi_1} \ln \pi_{\phi_1}^v(v) ds.
$$

$$
\tag{39}
$$

By substituting equation (39) into equation (37), the theorem is proven. The proof for $g_2(t, x; \phi_2, \bar{\phi}_1)$ follows a procedure analogous to that of $g_1(t, x; \bar{\phi}_2, \phi_1)$. $\qquad\square$

### B.5 Proof of Theorem 5

*Proof.* By algorithm 1, for time discretization $\delta t$, we have

$$\theta_{k+1} - \theta_k = \alpha_\theta \delta t^\beta \cdot \frac{1}{I} \sum_{i=0}^{I-1} \frac{1}{\delta t} \cdot$$

$$\left[ \frac{\partial J^\theta(t_k, x_k)}{\partial \theta} \left( J^\theta(t_{k+1}, x_{k+1}) - J^\theta(t_k, x_k) + r_k \delta t - \lambda \ln \pi(u_k \mid t_k, x_k) \delta t - \beta J^\theta(t_k, x_k) \delta t \right) \right].$$
$$(40)$$

According to Assumption 1, using Itô's fomula for $J^\theta$, we have

$$J^\theta(t_{k+1}, x_{k+1}) - J^\theta(t_k, x_k)$$
$$\approx J_t^\theta(t_k, x_k) \cdot \delta t + J_x^\theta(t_k, x_k) \cdot b(t_k, x_k, u_k) \cdot \delta t + J_x^\theta(t_k, x_k) \cdot \bar{\sigma}(t_k, x_k, u_k) \cdot W_{\delta t}$$
$$+ \frac{1}{2} J_{xx}^\theta(t_k, x_k) \cdot \bar{\sigma}^2(t_k, x_k, u_k) \cdot \delta t + o(\delta t)$$

where $W_{\delta t}$ denotes the Brownian motion over transitions Hereafter, we denote by $\tilde{C}$ a constant which is independent of $\delta t$. Then (40) becomes

$$\theta_{k+1} - \theta_k = \alpha_\theta \delta t^\beta \cdot \frac{1}{\delta t} \cdot \left[ \tilde{C}_1 \cdot \delta t + o(\delta t) \right].$$
$$(41)$$

(i) When $\beta = 1$, let $\delta t \longrightarrow 0$, we have

$$\frac{d\theta_t}{dt} = \tilde{C}(\theta_t, x_t, u_t).$$

Parameters in discrete-time convergence to continuous trajectory charactirized by well-posed ordinary differential equation.

(ii) When $\beta > 1$, let $\delta t \longrightarrow 0$, we have

$$\frac{d\theta_t}{dt} = \tilde{C}(\theta_t, x_t, u_t) \cdot (dt)^{\beta-1}.$$

Parameters in discrete-time convergence to initial state $\theta_0$.

(iii) When $\beta < 1$, let $\delta t \longrightarrow 0$, we have

$$\frac{d\theta_t}{dt} = \frac{\tilde{C}(\theta_t, x_t, u_t)}{(dt)^{1-\beta}}.$$

Parameters in discrete-time can reach infinity. $\qquad \square$

## C  Off-policy approach

Compared to the $q$-framework proposed in Jia and Zhou [2023], two limitations arise. First, constraints (27) or (21) are not easily verifiable in general environments. In particular, when the normalizing constant in the Gibbs measure is unavailable, it renders methods relying solely on a $q$-network without a policy network infeasible (Algorithms 1–3). Second, the policy-network-maintaining method introduced in Jia and Zhou [2023] remains an on-policy approach. And the proposed learning method in Jia and Zhou [2023]:

$$\phi \leftarrow \phi - \gamma \alpha_\phi dt \left[ \log \boldsymbol{\pi}^\phi \left( a_t^{\pi^\phi} \mid t, X_t \right) - \frac{1}{\gamma} q \left( t, X_t, a_t^{\pi^\phi}; \boldsymbol{\pi}^\phi \right) \right] \frac{\partial}{\partial \phi} \log \boldsymbol{\pi}^\phi \left( a_t^{\pi^\phi} \mid t, X_t \right) \quad (42)$$

introduces bias in off-policy settings. This leads to the accumulation of biases at each update step throughout the training.

The method proposed in this paper is a practical continuous-time off-policy algorithm. The approach ensures unbiased gradient estimation by policy network and reparameterized sampling during the policy gradient steps. From the update formula (19), it can be observed that in policy gradient update step, the only component related to the interaction trajectory is the martingale difference term.

As stated in Theorem 1, when the martingale property is ensured, the policy gradient update step becomes consistent with the on-policy estimation in Theorem 3. We emphasize that the martingale merely serves as a bridge in our approach to reinforcement learning and we approximately ensure the martingale property of the process $M_s$ via (19) instead of enforcing it. As mentioned in Section 3, we can choose whether to adopt importance sampling based on the specific problem, as it is not mandatory. The update method (19) effectively balances computational efficiency with mitigating the effects of distribution shift. And

$$\mathbb{E}_{\pi_\phi}\left[\lambda\partial_\phi\ln\left(\pi_\phi\left(u_{t_i}^\phi\mid t_i,x_{t_i}\right)\right)\delta t\right]=0,\tag{43}$$

we can taking the martingale difference term $\delta M$ as the advantage function, the update rule of policy $\pi^\phi$ aligns with discrete-time formulations.

## C.1  Comparative experiment

To facilitate a comparative analysis of the methods, we adopt the linear quadratic problem presented in Jia and Zhou [2023] as a benchmark. [8] The system dynamics are governed by matrix

$$dX_t^\pi=(AX_t^\pi+Ba_t^\pi)\,dt+(CX_t^\pi+Da_t^\pi)\,dW_t,\quad X_0^\pi=x_0,$$

and the objective is to maximize the payoff

$$\liminf_{T\to\infty}\frac{1}{T}\mathbb{E}\left[\int_0^T-\left(\frac{M}{2}(X_t^\pi)^2+RX_t^\pi a_t^\pi+\frac{N}{2}(a_t^\pi)^2+PX_t^\pi+Qa_t^\pi\right)\right.$$
$$\left.-\lambda\log\pi\left(a_t^\pi\mid X_t^\pi\right)dt\mid X_0^\pi=x_0\right].$$

In our simulation, the system parameters are set as $A=-1,B=C=0,D=1$, and the initial state $x_0=0$. The cost function parameters are $M=N=Q=2,R=P=1$, with temperature $\lambda=0.1$. The target policy, associated with parameterized q-function, is defined as $\pi(\cdot\mid x)=N(\psi_1x+\psi_2,\lambda e^{\psi_3})$ in Jia and Zhou [2023] and we approximate this target using a parameterized policy $\pi(\cdot\mid x)=N(\phi_1x+\phi_2,e^{\phi_3})$. The experiments use a time discretization step of $\delta t=0.1$. Each of the learning algorithms is run for a sufficiently long duration of $T=10^6$ steps.

The behavior policy is set to a normal distribution $N(-x-1,1)$:

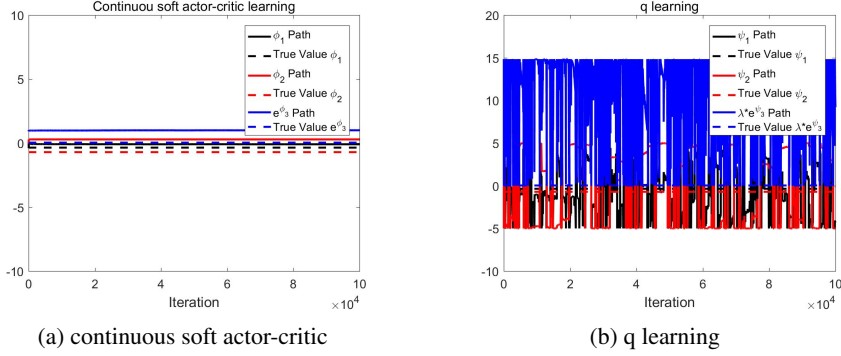

(a) continuous soft actor-critic          (b) q learning

Figure 3: Comparison of learning curves Using behavior policy $N(-x-1,1)$.

The behavior policy is set to a normal distribution $N(x+1,1)$:

---

[8]The repository includes code accompanied by implementation notes: `https://github.com/hh11813/continuous-soft-actor-critic`

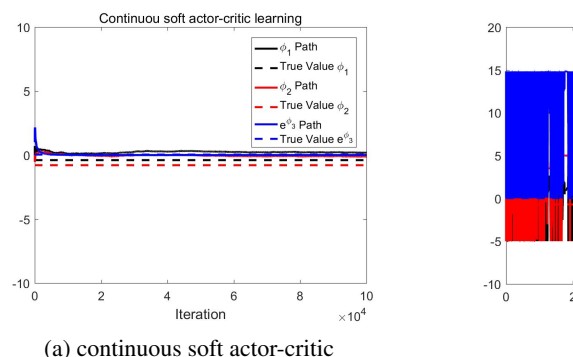

| (a) continuous soft actor-critic | (b) q learning |
|:---:|:---:|

Figure 4: Comparison of learning curves using behavior policy $N(x + 1, 1)$.

We can observe that the other method collapse, while our approach remains stable. This reflects the bias in learning results caused by on-policy method when there is a difference between the behavior policy and the target policy. The presence of the upper and lower bounds on the parameter curve in Figure 3 and Figure 4 is a result of the parameter clipping step incorporated into learning process in Jia and Zhou [2023].

# D   Plots and tables

## D.1   Gradients

Gradient explosion can be effectively addressed through gradient clipping. We focus on the vanishing gradient problem in this discussion. Tables 6 and 7 illustrate the gradient changes during the update process with seed $\{0\}$.

## D.2   Results

**Navigation**   (i) We compare the performance of multiple algorithms under varying time discretization parameters $\delta t$, with the baseline hyperparameter discount factor set to $\gamma = 0.9$. Results presented in Tables 4 and 8 consistently highlight the sensitivity to $\delta t$, indicating a lack of robustness. The experiments reported in Tables 2, 3 and 1 employ $3 \times 10^5$ frames, while Table 4 use $1.2 \times 10^5$ frames, all with random seeds $\{0, 1, 2\}$.

Table 8: Aggregate scores under navigation

| $\delta t$=0.1 | QMIX | IQL | MAPPO | MASAC |
|---|---|---|---|---|
| Median | 0.97 [0.97, 0.97] | 1.0 [1.0, 1.0] | 0.98 [0.98, 0.98] | 0.97 [0.97, 0.97] |
| IQM | 0.97 [0.97, 0.97] | 1.0 [1.0, 1.0] | 0.98 [0.98, 0.98] | 0.97 [0.97, 0.97] |
| Mean | 0.97 [0.97, 0.97] | 1.0 [1.0, 1.0] | 0.98 [0.98, 0.98] | 0.97 [0.97, 0.97] |
| Optimality Gap | 0.03 [0.03, 0.03] | 0.0 [0.0, 0.0] | 0.02 [0.02, 0.02] | 0.03 [0.03, 0.03] |
| $\delta t$=0.01 | QMIX | IQL | MAPPO | MASAC |
| Median | 0.63 [0.57, 0.67] | 0.61 [0.6, 0.62] | 0.55 [0.54, 0.57] | 0.55 [0.54, 0.56] |
| IQM | 0.63 [0.57, 0.67] | 0.61 [0.6, 0.62] | 0.55 [0.54, 0.57] | 0.55 [0.54, 0.56] |
| Mean | 0.63 [0.57, 0.67] | 0.61 [0.6, 0.62] | 0.55 [0.54, 0.57] | 0.55 [0.54, 0.56] |
| Optimality Gap | 0.37 [0.33, 0.43] | 0.39 [0.38, 0.4] | 0.45 [0.43, 0.46] | 0.45 [0.44, 0.46] |
| $\delta t$ | MADDPG (0.1) | MADDPG (0.01) | | |
| Median | 0.91 [0.88, 0.93] | 0.7 [0.52, 0.89] | | |
| IQM | 0.91 [0.88, 0.93] | 0.7 [0.52, 0.89] | | |
| Mean | 0.91 [0.88, 0.93] | 0.7 [0.52, 0.89] | | |
| Optimality Gap | 0.09 [0.07, 0.12] | 0.3 [0.11, 0.48] | | |

Table 6: Gradients of MADDPG (critic network) under sampling

| $\delta t$=0.1 | $\delta t$=0.01 |
|---|---|
| mlp.params.4.bias: 0.0382 | mlp.params.4.bias: 0.0007 |
| mlp.params.4.weight: 0.1038 | mlp.params.4.weight: 0.0047 |
| mlp.params.2.bias: 0.0237 | mlp.params.2.bias: 0.0004 |
| mlp.params.2.weight: 0.0594 | mlp.params.2.weight: 0.0043 |
| mlp.params.0.bias: 0.0353 | mlp.params.0.bias: 0.0003 |
| mlp.params.0.weight: 0.0793 | mlp.params.0.weight: 0.0032 |
| mlp.params.4.bias: 0.0260 | mlp.params.4.bias: 0.0108 |
| mlp.params.4.weight: 0.0797 | mlp.params.4.weight: 0.0172 |
| mlp.params.2.bias: 0.0161 | mlp.params.2.bias: 0.0058 |
| mlp.params.2.weight: 0.0444 | mlp.params.2.weight: 0.0156 |
| mlp.params.0.bias: 0.0241 | mlp.params.0.bias: 0.0042 |
| mlp.params.0.weight: 0.0571 | mlp.params.0.weight: 0.0128 |
| mlp.params.4.bias: 0.0190 | mlp.params.4.bias: 0.0133 |
| mlp.params.4.weight: 0.0502 | mlp.params.4.weight: 0.0193 |
| mlp.params.2.bias: 0.0123 | mlp.params.2.bias: 0.0071 |
| mlp.params.2.weight: 0.0522 | mlp.params.2.weight: 0.0170 |
| mlp.params.0.bias: 0.0191 | mlp.params.0.bias: 0.0051 |
| mlp.params.0.weight: 0.0910 | mlp.params.0.weight: 0.0141 |
| mlp.params.4.bias: 0.0188 | mlp.params.4.bias: 0.0076 |
| mlp.params.4.weight: 0.0383 | mlp.params.4.weight: 0.0165 |
| mlp.params.2.bias: 0.0120 | mlp.params.2.bias: 0.0041 |
| mlp.params.2.weight: 0.0453 | mlp.params.2.weight: 0.0153 |
| mlp.params.0.bias: 0.0184 | mlp.params.0.bias: 0.0029 |
| mlp.params.0.weight: 0.0828 | mlp.params.0.weight: 0.0110 |

Table 7: Gradients of CMASAC (critic network) under navigation ($\delta t$=0.01)

| CMASAC | TEST |
|---|---|
| mlp.params.4.bias: 0.1544 | mlp.params.4.bias: 0.2554 |
| mlp.params.4.weight: 2.7737 | mlp.params.4.weight: 4.0866 |
| mlp.params.2.bias: 0.0648 | mlp.params.2.bias: 0.0002 |
| mlp.params.2.weight: 0.1422 | mlp.params.2.weight: 0.0033 |
| mlp.params.0.bias: 0.7261 | mlp.params.0.bias: 0.0040 |
| mlp.params.0.weight: 0.9115 | mlp.params.0.weight: 0.0447 |
| mlp.params.4.bias: 0.5774 | mlp.params.4.bias: 0.5774 |
| mlp.params.4.weight: 9.1671 | mlp.params.4.weight: 9.2355 |
| mlp.params.2.bias: 0.0168 | mlp.params.2.bias: 0.0012 |
| mlp.params.2.weight: 0.0744 | mlp.params.2.weight: 0.0059 |
| mlp.params.0.bias: 0.1725 | mlp.params.0.bias: 0.0181 |
| mlp.params.0.weight: 0.4351 | mlp.params.0.weight: 0.0448 |
| mlp.params.4.bias: 0.2314 | mlp.params.4.bias: 0.2569 |
| mlp.params.4.weight: 3.9341 | mlp.params.4.weight: 4.1117 |
| mlp.params.2.bias: 0.0493 | mlp.params.2.bias: 0.0006 |
| mlp.params.2.weight: 0.1036 | mlp.params.2.weight: 0.0043 |
| mlp.params.0.bias: 0.5558 | mlp.params.0.bias: 0.0106 |
| mlp.params.0.weight: 0.6696 | mlp.params.0.weight: 0.0666 |

Table 9: Aggregate scores under navigation

|  | CMASAC |
| --- | --- |
| Median | 0.96 [0.93, 0.99] |
| IQM | 0.96 [0.92, 1.0] |
| Mean | 0.96 [0.93, 0.99] |
| Optimality Gap | 0.04 [0.01, 0.07] |

(ii) By increasing the number of random seeds to four and using more frames ($6 \times 10^5$), we explore the performance of the proposed algorithm under $\delta t = 0.01$. Comparison with the baseline results in Table 2 demonstrates that variations in extraneous factors such as random seeds did not have a decisive impact on the method's performance.

(iii) To separately examine the effects of the martingale orthogonality condition and parameter scaling, we compare the performance of the non-scaling CMASAC algorithm (TEST) and the MASAC algorithm in a *navigation* task using $6 \times 10^5$ frames, with random seeds $\{0, 1, 2\}$ and $\delta t = 0.01$. Table 10 and Figure 5, 6 present the experimental results. Table 10 indicates that the martingale orthogonality condition enhances the performance of the algorithm when $\delta t$ is small.

Table 10: Aggregate scores under navigation

|  | TEST |
| --- | --- |
| Median | 0.96 [0.93, 1.0] |
| IQM | 0.96 [0.93, 1.0] |
| Mean | 0.96 [0.93, 1.0] |
| Optimality Gap | 0.04 [0.0, 0.07] |

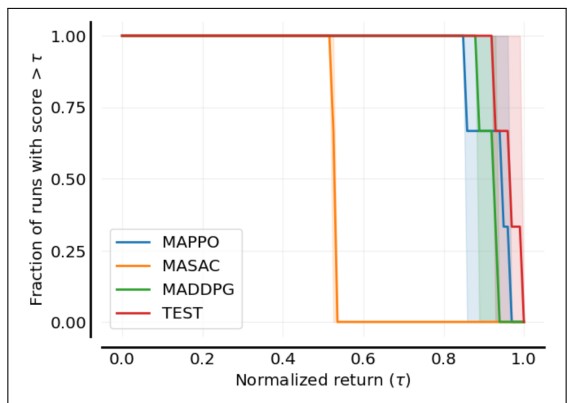

Figure 5: Performance profile for navigation task

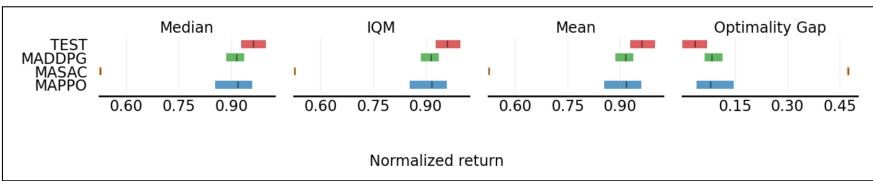

Figure 6: Aggregate score performance for navigation task

The TEST algorithm updates parameters according to modified version of Equation (19) in Algorithm 1:

$$\Delta\theta = \sum_{i=0}^{I-1} \frac{1}{I} \partial_\theta J^\theta\left(t_i, x_{t_i}\right) \delta M,$$

$$\Delta\phi = \sum_{i=0}^{I-1} \frac{1}{I}\left[\partial_\phi \ln \pi_{\phi_1}^v\left(v_{t_i}^\phi \mid t_i, x_{t_i}\right)\right]\delta M - \lambda\partial_\phi \ln\left(\pi_\phi^v\left(v_{t_i}^\phi \mid t_i, x_{t_i}\right)\right)\delta t,$$

with learning rates $\alpha_\theta$ and $\alpha_\phi$.

**Sampling**    (i) In Section 4 of the paper, we evaluate the performance of the MASAC and MADDPG algorithms in the *sampling* task. Here, we present results form additional algorithms.

Table 11: Aggregate scores under sampling

| $\delta t$=0.1 | QMIX | IQL | MAPPO |
|---|---|---|---|
| Median | 0.87 [0.75, 1.0] | 0.93 [0.89, 0.96] | 0.53 [0.45, 0.62] |
| IQM | 0.87 [0.75, 1.0] | 0.93 [0.89, 0.96] | 0.53 [0.45, 0.62] |
| Mean | 0.87 [0.75, 1.0] | 0.93 [0.89, 0.96] | 0.53 [0.45, 0.62] |
| Optimality Gap | 0.13 [0.0, 0.25] | 0.07 [0.04, 0.11] | 0.47 [0.38, 0.55] |
| $\delta t$=0.01 | QMIX | IQL | MAPPO |
| Median | 0.53 [0.46, 0.57] | 0.67 [0.46, 1.0] | 0.45 [0.29, 0.76] |
| IQM | 0.53 [0.46, 0.57] | 0.67 [0.46, 1.0] | 0.45 [0.29, 0.76] |
| Mean | 0.53 [0.46, 0.57] | 0.67 [0.46, 1.0] | 0.45 [0.29, 0.76] |
| Optimality Gap | 0.47 [0.43, 0.54] | 0.33 [0.0, 0.54] | 0.55 [0.24, 0.71] |

Similar to Table 8, when the discount factor is set to $\gamma = 0.9$, the results in Table 12 demonstrate that the algorithms exhibit unstable performance with respect to time discretization $\delta t$.

Table 12: Aggregate scores under sampling

| $\delta t$=0.1 | QMIX | IQL | MAPPO |
|---|---|---|---|
| Median | 0.93 [0.89, 0.98] | 0.93 [0.88, 1.0] | 0.54 [0.44, 0.59] |
| IQM | 0.93 [0.89, 0.98] | 0.93 [0.88, 1.0] | 0.54 [0.44, 0.59] |
| Mean | 0.93 [0.89, 0.98] | 0.93 [0.88, 1.0] | 0.54 [0.44, 0.59] |
| Optimality Gap | 0.07 [0.02, 0.11] | 0.07 [0.0, 0.12] | 0.46 [0.41, 0.56] |
| $\delta t$=0.01 | QMIX | IQL | MAPPO |
| Median | 0.62 [0.52, 0.72] | 0.62 [0.52, 0.71] | 0.51 [0.33, 0.86] |
| IQM | 0.62 [0.52, 0.72] | 0.62 [0.52, 0.71] | 0.51 [0.33, 0.86] |
| Mean | 0.62 [0.52, 0.72] | 0.62 [0.52, 0.71] | 0.51 [0.33, 0.86] |
| Optimality Gap | 0.38 [0.28, 0.48] | 0.38 [0.29, 0.48] | 0.49 [0.14, 0.67] |

For the *sampling* task with the discount factor $\gamma = 0.99$, CMASAC maintains superior performance under the time discretization $\delta t = 0.01$. As demonstrated by the consistent results in Table 13, adjustments to the discount factor do not compromise the robustness of CMASAC, highlighting its stability across varying hyperparameter configurations. Experiments in Tables 11, 12 and 13 were conducted using different numbers of frames ($6 \times 10^4$, $2.4 \times 10^5$).

(ii) For the extended evaluation, we test CMASAC with $3 \times 10^6$ frames and $\delta t = 0.01$ on *sampling*, the corresponding results are shown in Table 14 and 15. The unscaled continuous algorithm (TEST) demonstrated superior results compared to other methods (Table 15, Figures 7 and 8). Cross-referencing Tables 14 and 15 further validates that martingale orthogonality condition and hyperparameter scaling jointly contribute to performance improvements.

Table 13: Aggregate scores of CMASAC under sampling

|  | CMASAC |
| --- | --- |
| IQM | 0.95 [0.92, 1.0] |
| Mean | 0.95 [0.92, 1.0] |
| Median | 0.95 [0.92, 1.0] |
| Optimality Gap | 0.05 [0.0, 0.08] |

Table 14: Aggregate scores under sampling

|  | CMASAC |
| --- | --- |
| Median | 0.97 [0.94, 1.0] |
| IQM | 0.97 [0.94, 1.0] |
| Mean | 0.97 [0.94, 1.0] |
| Optimality Gap | 0.03 [0.0, 0.06] |

Table 15: Aggregate scores under sampling

|  | TEST | MASAC | MAPPO | MADDPG |
| --- | --- | --- | --- | --- |
| Median | 0.92 [0.84, 1.0] | 0.81 [0.8, 0.81] | 0.76 [0.72, 0.81] | 0.8 [0.77, 0.84] |
| IQM | 0.92 [0.84, 1.0] | 0.81 [0.8, 0.81] | 0.76 [0.72, 0.81] | 0.8 [0.77, 0.84] |
| Mean | 0.92 [0.84, 1.0] | 0.81 [0.8, 0.81] | 0.76 [0.72, 0.81] | 0.8 [0.77, 0.84] |
| Optimality Gap | 0.08 [0.0, 0.16] | 0.19 [0.19, 0.2] | 0.24 [0.19, 0.28] | 0.2 [0.16, 0.23] |

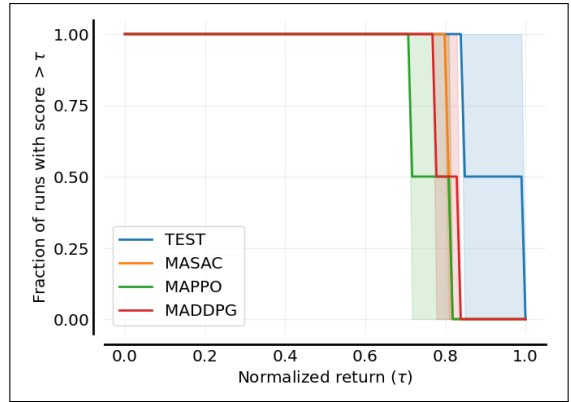

Figure 7: Performance profile for sampling task

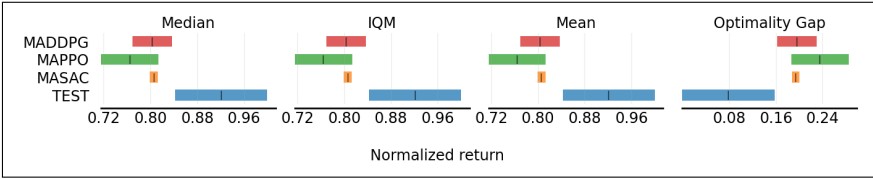

Figure 8: Aggregate score performance for sampling task

**Balance**  We evaluate the proposed algorithm in *balance* environment. The result in Table 16 demonstrates the potential of CMASAC in near-continuous-time settings. A comparison of the results in Tables 16 and 17 under $\delta t = 0.01$ reveals that both the martingale orthogonality condition and hyperparameter scaling contribute to improved performance and increased robustness of the

algorithms in near-continuous environments. Test results of the method in more scenarios remain to be explored.

Table 16: Aggregate scores of CMASAC under balance

| $\delta t$ | 0.01 |
|---|---|
| IQM | 0.94 [0.88, 1.0] |
| Mean | 0.94 [0.88, 1.0] |
| Median | 0.94 [0.88, 1.0] |
| Optimality Gap | 0.06 [0.0, 0.12] |

Table 17: Aggregate scores of TEST under balance

| $\delta t$ | 0.01 |
|---|---|
| IQM | 0.93 [0.89, 1.0] |
| Mean | 0.93 [0.89, 1.0] |
| Median | 0.93 [0.89, 1.0] |
| Optimality Gap | 0.07 [0.0, 0.11] |

### D.3 Significance

The experimental evaluation in this study was conducted using tool (based on Agarwal et al. [2021] and Gorsane et al. [2022]): `https://github.com/instadeepai/marl-eval`.

Considering the setting in which a reinforcement learning algorithm is evaluated on $M$ tasks and $N$ independent runs executed for each task. Then we derive normalized score $x_{m,n}, m = 1, \cdots, M$ and $n = 1, \cdots, N$. Here, we briefly describe the meaning of the experimental results tables and figures.

- IQM: Inter-quantile Mean
- Optimality Gap: Optimality Gap can be thought of as the how far an algorithm is from optimal performance at a given task.
- Confidence interval (CI): These provides an estimated possible range for an unknown value. We choose a 95 confidence interval in our experiments.
- Aggregate score performance: The confidence intervals shown alongside the point estimates (black bars) are the 95 stratified bootstrap confidence intervals.
- Performance profiles: The algorithm's normalized score on the $m^{\text{th}}$ task as a real-valued random variable $X_m$. Then, the score $x_{m,n}$ is a realization of the random variable $X_{m,n}$, which is identically distributed as $X_m$. For $\tau \in \mathbb{R}$, we define the tail distribution function of $X_m$ as $F_m(\tau) = P(X_m > \tau)$. For any collection of scores $y_{1:K}$, the empirical tail distribution function is given by $\hat{F}(\tau; y_{1:K}) = \frac{1}{K} \sum_{k=1}^{K} \mathbf{1}[y_k > \tau]$. In particular, we write $\hat{F}_m(\tau) = \hat{F}(\tau; x_{m,1:N})$. This explains the meaning of Figure 7 and the significance of the axes.

## E Implementation details

### E.1 Environments

Figures 1 and 9 illustrate several VMAS task scenarios.

- *Navigation* : Randomly spawned agents (circles with surrounding dots) need to navigate to randomly spawned goals (smaller circles). Agents need to use LIDARs (dots around them) to avoid running into each other. For each agent, we compute the difference in the relative distance to its goal over two consecutive timesteps. The mean of these values over all agents composes the shared reward, incentivizing agents to move towards their goals. Each agent observes its position, velocity, lidar readings, and relative position to its goal.

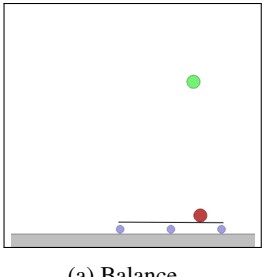

(a) Balance.

Figure 9: VMAS multi-robot control task–Balance–used in the experiments.

- *Sampling* : Agents are spawned randomly in a workspace with an underlying Gaussian density function composed of three Gaussian modes. Agents need to collect samples by moving in this field. The field is discretized to a grid (with agent-sized cells) and once an agent visits a cell its sample is collected without replacement and given as a reward to the whole team. Agents can use a lidar to sense each other in order to coordinate exploration. Apart from lidar, position, and velocity observations, each agent observes the values of samples in the 3x3 grid around it.

- *Balance* : Agents (blue circles) are spawned uniformly spaced out under a line upon which lies a spherical package (red circle). The team and the line are spawned at a random $x$ position at the bottom of the environment. The environment has vertical gravity. The relative $x$ position of the package on the line is random. In the top half of the environment, a goal (green circle) is spawned. The agents have to carry the package to the goal. Each agent receives the same reward which is proportional to the distance variation between the package and the goal over two consecutive timesteps. The team receives a negative reward of $-10$ for making the package or the line fall to the floor. The observations for each agent are: its position, velocity, relative position to the package, relative position to the line, relative position between package and goal, package velocity, line velocity, line angular velocity, and line rotation $\mod \pi$. The environment is done either when the package or the line falls or when the package touches the goal.

### E.2 Hyperparameters

**Random seeds**    We select random seeds $\{0, 1, 2\}$.

**Network architecture**    The policy and critic models are constructed with MLP layers, and the MLP architecture is defined as follows:

- num_cells: [256, 256]
- layer_class: torch.nn.Linear
- activation_class: torch.nn.Tanh

**Hyperparameters details**    Tables 18, 19 and 20 show configurations of different algorithms. These algorithm-specific hyperparameters take precedence over the common hyperparameters.

Table 20: Config details of selected algorithms

| QMIX |
| --- |
| delay_value: True |
| loss_function: "l2" |
| mixing_embed_dim: 32 |

And the shared parameters across all experimental algorithms are listed below:

Table 18: Config details of selected algorithms

| IQL | MASAC | MADDPG |
|---|---|---|
| | share_param_critic: True | share_param_critic: True |
| | num_qvalue_nets: 2 | |
| loss_function: "l2" | loss_function: "l2" | loss_function: "l2" |
| delay_value: True | delay_qvalue: True | delay_value: True |
| | target_entropy: "auto" | |
| | discrete_target_entropy_weight: 0.2 | |
| | alpha_init: 1.0 | |
| | min_alpha: null | |
| | max_alpha: null | |
| | fixed_alpha: False | |
| | scale_mapping: "biased_softplus_1.0" | |
| | use_tanh_normal: True | use_tanh_mapping: True |

Table 19: Config details of selected algorithms

| MAPPO | CMASAC |
|---|---|
| share_param_critic: True | share_param_critic: True |
| clip_epsilon: 0.2 | num_value_nets: 1 |
| entropy_coef: 0.0 | loss_function: "l2" |
| critic_coef: 1.0 | target_entropy: "auto" |
| loss_critic_type: "l2" | alpha_init: 1.0 |
| minibatch_advantage: False | min_alpha: null |
| | max_alpha: null |
| | fixed_alpha: False |
| scale_mapping: "biased_softplus_1.0" | scale_mapping: "biased_softplus_1.0" |
| use_tanh_normal: True | use_tanh_normal: True |
| lmbda: 0.9 | gamma: 0.9 |

- share_policy_params: True
- (discount factor) gamma: 0.99
- (learning rate) lr: 0.00005
- (adam optimizer) adam_eps: 0.000001
- (soft target update) polyak_tau: 0.005
- (initial epsilon for annealing) exploration_eps_init: 0.8
- (final epsilon after annealing) exploration_eps_end: 0.01
- max_n_frames: 3_000_000
- on_policy_collected_frames_per_batch: 6000
- on_policy_n_envs_per_worker: 10
- on_policy_n_minibatch_iters: 45
- on_policy_minibatch_size: 400
- off_policy_collected_frames_per_batch: 6000
- off_policy_n_envs_per_worker: 10
- off_policy_n_optimizer_steps: 1000
- off_policy_train_batch_size: 128
- off_policy_memory_size: 1_000_000
- off_policy_init_random_frames: 0
- off_policy_use_prioritized_replay_buffer: False

- evaluation_interval: 120_000
- evaluation_episodes: 10
- evaluation_deterministic_actions: False

**Evaluation details**

- Evaluation intervals: It refers to the fixed number of time steps, after which training is suspended, to be able to evaluate an algorithm for a fixed number of runs/epsiodes. The evaluation frequency must ideally be associated with a duration which we record as the evaluation duration.
- Number of independent evaluations per interval: This is the amount of evaluations that are performed at each evaluation interval.

**Computer resources**    The experiments are conducted on a system equipped with Intel Xeon Silver 4314 CPU (2.40GHz, 16 physical cores) and an NVIDIA RTX 4090 GPU (24GB VRAM). Each independent algorithm experiment run consumes about 1.5GB of CPU RAM and 2.2GB of GPU VRAM on average.

