# OpenReview forum: "Continuous Soft Actor-Critic: An Off-Policy Learning Method Robust to Time Discretization"
_NeurIPS.cc/2025/Conference — NeurIPS 2025 poster_

### Official Review · Reviewer_U79M · 2025-06-22

**Clarity:** 2
**Significance:** 2
**Originality:** 2
**Rating:** 4
**Confidence:** 3

**Summary:**

This paper proposes Continuous Soft Actor-Critic (CSAC), an off-policy reinforcement learning algorithm formulated in continuous time control. It claims to overcome the limitations of existing discrete-time DRL algorithms (such as SAC, DDPG, MAPPO), whose performance may suffer as the control frequency increases (i.e., as the time discretization step δt → 0). The authors extend this framework to a multi-agent setting (CMASAC) and introduce policy evaluation methods based on martingale orthogonality conditions derived from stochastic control theory. The paper provides theoretical derivations and some experimental comparisons in VMAS environments.

**Questions:**

See weaknesses.

**Ethical Concerns:**

["NO or VERY MINOR ethics concerns only"]

**Final Justification:**

I sincerely appreciate the authors' clarification and references. It seems that considering such special formulation of financial market applications is feasible. I read all the rebuttal content and acknowledge the technical completeness of this article, which is why I am willing to improve my score. However, I still don't think this is a major factor in current MARL research. I reserve my opinion on the experimental settings and effects. The two tasks of VMAS are too simple, and this assumption are inadequate to support the applicability of a large number of non-Gaussian policy task scenarios such as SMAC, GRF, OverCooked, etc. After considering other reviewers and AC's opinions, I would like to keep my final score as 4.

**Limitations:**

Yes.

**Paper Formatting Concerns:**

No problems.

**Quality:**

2

**Strengths And Weaknesses:**

Strengths
1. Theoretical derivations are mostly well-structured for the single-agent case and rooted in established stochastic analysis.
2. Recognition of time-dependent hyperparameter scaling is interesting.
3. The critique of TD-based loss in stochastic continuous systems is an important and underexplored technical consideration.

Weaknesses
1. While the paper builds upon recent literature on continuous-time RL (e.g., Doya 2000, Tallec et al. 2019, Jia and Zhou 2022), its main contribution — CSAC — appears to be a direct and relatively incremental extension of prior theoretical work (especially Jia and Zhou 2022). Core ideas like martingale-based policy evaluation, learning rate scaling, and entropy-regularized control appear to follow standard stochastic control theory.
2. Time discretization in RL is a modeling choice naturally defined in Markov Decision Process, and existing algorithms like PPO or SAC are known to work well under fine-grained simulations (e.g., MuJoCo, IsaacGym). The authors need to clarify why advancing continuous-time theory is practically relevant in the modern RL context and is helpful for any real applications? The paper doesn’t convincingly identify a problem that fundamentally requires continuous-time methods.
3. Core theoretical results begin with the single-agent case, yet no single-agent experiments are performed. It will be helpful to support the theoretical proof to compare CSAC with modified standard baseline algorithms. The jump to multi-agent settings lacks a coherent rationale, and adaptations of the theory to limited observability (POMDP assumption in general MARL research) are not discussed or validated.
4. The experimental setup is inadequate.

First, MARL baselines such as MADDPG and MASAC perform poorly under small δt, possibly due to unnormalized rewards and unscaled learning rates. In VMAS, reward functions are typically based on the distance between agents and landmarks. As the control time interval δt decreases, per-step rewards naturally scale down linearly, which in turn reduces the magnitude of policy gradients — as illustrated in Table 4. The authors should normalize rewards and adjust baseline hyperparameters accordingly to ensure fair and meaningful comparisons.

Second, the training curves of algorithms are not provided. The experiments are conducted on 3 random seeds, it is suggested to conduct all experiments on 5 seeds as previous RL literature indicated. Besides, across all experiments, the reported IQM, Median, and Mean values are nearly identical, sometimes even exactly the same. This is highly unusual for reinforcement learning experiments, especially in multi-agent settings, where result variance (due to stochastic environments and policy instability) is typically high.

5. The manuscript suffers from poor writing. The introduction fails to provide sufficient background and a clear problem statement, and the literature review is severely lacking in breadth and depth. Many necessary references are missed in papers such as PPO, SAC, CTDE, IQL.

---

> ### Author Rebuttal · Authors · 2025-07-30
>
> We appreciate your questions and comments. Please find our responses to your questions below:
>
> 1. (1)Regarding comparisons with other works, we emphasize that Doya [2000] and Tallec et al. [2019] are discussed in deterministic systems, which leads to fundamental differences (section 3.1). As we stated in the paper, in continuous stochastic environments such as (1), designing the loss function as MSTDE no longer guarantees a reduction in the function approximation error, but only helps to reduce the fluctuations of the learned process (true value function preserves the martingale property, and the corresponding MSTDE is not 0). Therefore, algorithms that use MSTDE exhibit differences in the continuous-time and require re-examination.
>
> (2)As for Jia and Zhou [2022], as we have stated in the paper, it is mainly focuses on on-policy framework. In contrast, we propose an off-policy method with Theorem 1 serving as its theoretical foundation. To more intuitively illustrate the differences, we conduct two sets of comparative experiments between our method and the approach proposed by Jia and Zhou [2022, 2023], using the example from Jia and Zhou [2023, Section 7.2] and the code provided by them. We set the behavior policies as normal distributions with mean parameters of (0.5, 0.5) and (1, 1), respectively, and a variance of 1 for both. We observe that the method from Jia and Zhou [2022, 2023] collapse (oscillatory collapse and significant deviation from the true value​), while our approach remains stable (once the update is approved by the official, we will release the code and figures). We simply list the learning error ($\phi_2-\phi_2^{true}$) of the parameter $\phi_2$ when the behaviour policy is set to $N(x+1,1)$.
>
> |iteration(10^4)|2|4|6|8|10|
> |------------|------------------|------------------|------------------|------------------|------------------|
> |CSAC| 0.706|0.780|0.701|0.683|0.670|
> |q learning(Jia and Zhou,2023)|-0.586|-4.211|5.756|-4.053|-3.664|
> |continuous TD(0)(Jia and Zhou,2022)|-4.149|-4.142|-4.139|-4.112|-4.098|
>
> This also serves as a validation of the feasibility of our method in the single-agent setting. Moreover, there are still some issues in approaches proposed by Jia and Zhou [2022] that require further investigation. For instance, it is not straightforward to implement the gradient update derived in Jia and Zhou [2022] on a dynamic computation graph to perform backpropagation. Issues such as parameter detach, test function selection, and the non-differentiability of sampling make it difficult to apply directly, which limits its effectiveness in handling high-dimensional problem or large-scale datasets.
>
> 2. Although time discretization is essential for implementing RL algorithms, we would like to emphasize the following points:
>
> (1) For scenarios involving persistent noise (noise that resembles Brownian motion or more intuitively, there are many spikes visible in the plotted data), the continuous-time framework proposed in this paper should be considered (statistical characteristics are unknown, requiring diverse verification methods). As discussed in response to Point 1(1), employing existing mean-square error formulations may lead to incorrect results in such settings. Moreover, the SDE framework is a more general paradigm, inherently accommodating properties such as state continuity and Markovian dynamics, while also offering enhanced extensibility (e.g. non-Markovian scenarios). High-frequency financial data and high-dimensional stochastic control problems are typical examples of such framework. The following are current or potential applications.
>
> [5] H. Zhao, H. Chen, J. Zhang, et al. Score as action: Fine-tuning diffusion generative models by continuous-time reinforcement learning. arXiv preprint arXiv:2502.01819, 2025.
>
> [6] H. Wang, X. Y. Zhou. Continuous‐time mean–variance portfolio selection: A reinforcement learning framework. Mathematical Finance, 2020, 30(4): 1273-1308.
>
> [7] C. Marinelli. The stochastic goodwill problem. European Journal of Operational Research, 2007, 176(1): 389-404.
>
> (2) For simulation environments like MuJoCo, IsaacGym, as demonstrated by Tallec et al. [2019], Q-learning collapses under smaller time discretizations than used in default simulation settings. We argue that existing algorithms, such as PPO or SAC, do not offer guaranteed robustness at ultra-small time scales. With advances in sensor technology, increased interaction data makes further exploration possible. Considering the connection between the martingale condition and existing traditional algorithms (see pages 22-25 of Jia and Zhou [2022a]), along with the study on hyperparameter scaling, we can expect the proposed algorithm to perform well in a broader range of near-continuous scenarios (especially in cases that employ a Gaussian distribution policy like this paper). For instance, in the VMAS environment we selected (though the statistical properties are uncertain), the experimental results indicate that the method proposed in this paper is suitable for such scenarios. Therefore, we believe that the problem investigated in this paper is timely and necessary.
>
> 3. We thank the reviewer for the comments regarding the experiments. In the final draft, we will include the experiment mentioned in Point 1, which will not only validate the feasibility of our method in the single-agent setting but also allow for comparisons with other approaches. Our discussion of the multi-agent scenario is built upon the framework defined in Section 2, which is ​​consistent and coherent​​ with our research in the single-agent setting. We agree that continuous-time multi-agent systems remain an area in need of further exploration, and this work can be viewed as a preliminary investigation in this domain. The field of continuous-time multi-agent systems is open, with very few existing studies addressing its challenges. In future research, we will delve deeper into areas such as the ​​partial observability, as suggested by reviewers.
>
> 4. (1)In this paper, the experimental setup is based on the fact that VMAS includes multiple different scenarios, with different reward definitions. The reward function is not necessarily linear with respect to the time discretization step, as in the case of the sampling scenario. In the navigation and balance scenarios selected in the paper, rewards are also generated by events such as landing, collisions between agents, and reaching the target position, in addition to changes in displacement. Therefore, we did not choose to directly apply reward scaling to algorithms. As can also be seen from the performance of DDPG in the figures on page 17 of Tallec et al. [2019], scaling rewards and hyperparameters across different $\delta t$ does not necessarily improve performance. However, we agree with the reviewer's point that this factor should be taken into consideration. We have added experiments with five random seeds ${0,1,2,3,4}$ and include hyperparameters (learning rate, discount factor) and reward scaling. The results do not appear to contradict the conclusions of the paper. Among them, QMIX, and MASAC exhibit performance degradation, IQL, MAPPO and MADDPG show improvement. Except for MADDPG, the performance of algorithms remains inferior to that achieved with $\delta t=0.1$. Reward scaling seems insufficient to offset the effects caused by reducing the time scale(experiments and results will be released upon official approval).
>
> ||QMIX|IQL | MAPPO|MASAC|MADDPG
> |------------|------------------|------------------|------------------|------------------|------------------|
> Median|0.75 [0.69, 0.81]|0.92 [0.89, 0.94]|0.87 [0.87, 0.89]|0.49 [0.48, 0.5]|0.88 [0.79, 0.96]
> IQM|0.74 [0.67, 0.83]|0.92 [0.88, 0.95]|0.87 [0.86, 0.89]|0.49 [0.47, 0.5]|0.9 [0.76, 0.98]
> Mean|0.75 [0.69, 0.81]|0.92 [0.89, 0.94]|0.87 [0.87, 0.89]|0.49 [0.48, 0.5]|0.88 [0.79, 0.96]
> Optimality Gap|0.25 [0.19, 0.31]|0.08 [0.06, 0.11]|0.13 [0.11, 0.13]|0.51 [0.5, 0.52]|0.12 [0.04, 0.21]
>
> (2)We understand the reviewer’s concerns regarding the experimental results. However, we would like to refer to Figure 4 on page 7 of the paper Matteo Bettini et al. [2024](BenchMARL). As shown in the figure, the mean of the baseline algorithms used in the comparison, along with the IQM and median values, are either the same or very close. Therefore, this is not an issue caused by our proposed method or the implementation of the code, but rather a characteristic.
>
> 5. We thank the reviewer for the suggestion regarding the references. We have already provided a comparative background for our method with respect to existing approaches in Lines from 42 to 44 and 70 to 76. In various parts of the paper, such as Lines 28, 56, and 68, we have cited relevant references to the algorithms related to our study (CTDE was also introduced by Lowe et al. [2017]). We appreciate the reviewer’s comment on the missing reference to the PPO method, and we will include additional references in the final draft to improve readability.
>
> [1] J. Schulman, F. Wolski, P. Dhariwal, A. Radford, and O. Klimov. Proximal policy optimization algorithms. arXiv preprint arXiv:1707.06347, 2017.
>
> Other references mentioned in the reply are listed below.
>
> [2] M. Bettini, A. Prorok, V. Moens. Benchmarl: Benchmarking multi-agent reinforcement learning. Journal of Machine Learning Research, 2024, 25(217): 1-10.
>
> [3] Y. Jia, X. Y. Zhou. q-Learning in continuous time[J]. Journal of Machine Learning Research, 2023, 24(161): 1-61.
>
> [4] Y. Jia, X. Y. Zhou. Policy gradient and actor-critic learning in continuous time and space: Theory and algorithms[J]. Journal of Machine Learning Research, 2022, 23(275): 1-50.

---

> > ### Comment · Reviewer_U79M · 2025-08-05
> >
> > I appreciate the author's response to questions and the efforts to conduct additional experiments, which partially resolved my concerns and increased the reliability of the article.
> >
> > However, I still keep concerns about the adaptability of the research content to the current advanced MARL task scenarios, and the impact of the research topic on the MARL community, especially practical significance. In particular, the authors mentioned that the continuous-time framework of this paper is very suitable for scenarios with Brownian motion-like noise, as well as high-frequency financial data and high-dimensional stochastic control. However, I am very curious about how high-frequency financial data can be modeled as a MARL task, and whether such a simple task like VMAS has such noise characteristics? I am willing to raise my score to 3.

---

> > > ### Author Response · Authors · 2025-08-06
> > >
> > > We appreciate the reviewer's comments. Please find our responses below.
> > >
> > > 1. Here, we list two scenarios where buyers and sellers can be modeled using Multi-Agent Reinforcement Learning, along with relevant references:
> > >
> > > (1) Portfolio Management: MARL can be employed to optimize investment portfolios, where stock prices are modeled by geometric Brownian motion. Each agent represents a trader, with different traders adopting distinct trading strategies to diversify risk and pursue profits. The policy network outputs the quantity of assets to trade, the reward function is defined based on returns and risk preferences, and the wealth process follows the formulation described in this paper. For this scenario, please refer to [1].
> > >
> > > [1] Kim, R., Kang, J., Lee, J., & Yi, S. W. (2020, July). MAPS: Multi-Agent reinforcement learning-based Portfolio management System. In Proceedings of the Twenty-Ninth International Joint Conference on Artificial Intelligence. International Joint Conferences on Artificial Intelligence Organization.
> > >
> > > (2) Market Making: MARL can also be used to simulate market maker behavior in financial markets. In this setting, agents represent market makers and investors. The market maker's policy outputs bid and ask quote parameters, with the reward function incorporating spread profits and inventory risk. The investor agent's policy determines trading strategies. The mid-price, reference spread, market maker inventory, and investor positions serve as state variables, following the framework described in the paper. For this scenario, please refer to [2].
> > >
> > > [2] Ganesh, S., Vadori, N., Xu, M., Zheng, H., Reddy, P., & Veloso, M. (2019). Reinforcement learning for market making in a multi-agent dealer market. arXiv preprint arXiv:1911.05892.
> > >
> > > In both of these scenarios, agents are trained on historical interaction data.
> > >
> > > 2. For a given scenario, even if the dynamics follow deterministic physical laws, due to the use of Gaussian policies in many algorithms, the system dynamics can still be represented in the form of a stochastic differential equation as the time step approaches zero. Therefore, it is natural to test the algorithm in environments such as VMAS (composite noise from random initialization, interaction noise, and sensor noise, exhibiting Gaussian characteristics) or other simple tasks (where the noise lacks a specific distribution).
> > >
> > > We hope our responses has been helpful in addressing your questions regarding the algorithm's application.

---

> > > > ### Comment · Reviewer_U79M · 2025-08-07
> > > >
> > > > I sincerely appreciate the authors' clarification and references. It seems that considering such special formulation of financial market applications is feasible. I read all the rebuttal content and acknowledge the technical completeness of this article, which is why I am willing to improve my score. However, I still don't think this is a major factor in current MARL research. I reserve my opinion on the experimental settings and effects. The two tasks of VMAS are too simple, and this assumption are inadequate to support the applicability of a large number of non-Gaussian policy task scenarios such as SMAC, GRF, OverCooked, etc. From this perspective, I would like to keep my final score as 3.

---

### Official Review · Reviewer_VtE3 · 2025-07-01

**Clarity:** 2
**Significance:** 3
**Originality:** 3
**Rating:** 4
**Confidence:** 4

**Summary:**

This paper present a novel off-policy reinforcement learning algorithm, called Continuous Soft Actor-Critic (CSAC), for stochastic continuous-time learning. The key observation of this paper is the enforcement of the martingale property in value functions, along with carefully designed gradient and hyperparameter scaling laws, resulting in a δt-robust parameter update rule. Empirical evaluations on some benchmarks validate the algorithm’s robustness to time discretization.

**Questions:**

1. What's the definition of the continuous-time setting?

2. How is the value function constraint given by Equation 13 implemented during policy training?

3. Does alternating training promote coordination between agents?

4. Does the single-agent martingale orthogonality condition also hold in multi-agent settings?

**Ethical Concerns:**

["NO or VERY MINOR ethics concerns only"]

**Final Justification:**

We appreciate the authors' efforts in addressing my concerns.

**Limitations:**

yes

**Paper Formatting Concerns:**

Some equations lack numbers.

**Quality:**

3

**Strengths And Weaknesses:**

Strengths:

1. This paper extends Soft Actor-Critic from the discrete-time setting to the continuous-time setting. The proposed Continuous Soft Actor-Critic is robust to environment time discretization and achieves better empirical performance.

2. The paper introduces a novel martingale-theory-based policy evaluation method to approximate true value functions in continuous-time settings.

3. A new approach is proposed for adjusting the algorithm’s hyperparameters in stochastic continuous-time environments to accommodate changes in the time step δt.

Weaknesses:

1. The authors should formally define the continuous-time setting in the Preliminaries section.

2. The authors claim that minimizing MSTDE to learn a parameterized function cannot guarantee convergence to the value function in stochastic continuous-time environments. Instead, they propose leveraging the martingale property of the value function and enforce constraints on the parameterized function to preserve this property based on Equation 13. However, it is unclear how this constraint is implemented during policy evaluation.

3. More details about the Soft Actor-Critic algorithm should be provided. The multi-agent framework assumes two independently learning agents, and the alternating training may fail to capture coordination between them. Centralized training requires more global information about both agents.

4. It is questionable whether the single-agent martingale orthogonality condition also holds in multi-agent settings, as enforcing martingale orthogonality individually may not guarantee convergence to the joint multi-agent value function. More theoretical justification in the multi-agent case is needed.

---

> ### Author Rebuttal · Authors · 2025-07-29
>
> Thank you for your valuable comments. Please find our responses to your questions below:
>
> 1. In this paper, "continuous-time" refers to the use of continuous-time modeling in theory; that is, the states, actions, and environment satisfy the definitions provided in Section 2. We only apply uniform time discretization during the algorithm and execution phases. This paper aims to provide a solution for scenarios involving persistent noise (similar to Brownian motion, or more intuitively, for which many spikes are visible in the state trajectories). Given that we typically only have access to data without knowing the statistical characteristics of the noise, the continuous-time framework and the methodology proposed in this paper should be included for comparison in practice.
>
> 2. Although Equation (13) in this paper represents a moment condition, solving for the parameters in (13)—whether through a stochastic approximation method or a generalized method of moments (GMM), results in an iterative update rule that is formally identical to semi-gradient optimization (see pages 24,25 of Jia and Zhou [2022a] and Sutton [2008]). Therefore, the optimization approach we adopt is outlined in Algorithm 1: we use the gradient of the parameters $\theta$ along with the Adam optimizer to update them. In the experiments, we employed the 'distance\_loss' and 'l2\_loss' functions from torchrl as the loss functions for the updates, with the gradient forms given by Equation (17).
>
> 3. Regarding multi-agent systems, discussion in this paper is confined to the framework established in Section 2, where we assume fixed policy for other agents. As formalized in Equation (14), our theoretical analysis postulates that agent v possesses prior knowledge of policies from other agents u (reward takes the expectation with respect to $\pi^u$). Consequently, conclusions derived for single-agent scenarios can be extended to this multi-agent configuration (Theorem 2 on page 5; Theorem 5 on page 18). In our experiments, we try to implement an approach closely aligned with the theoretical framework (alternating training). We acknowledge that the theory of multi-agent reinforcement learning requires further exploration, and our framework cannot cover broad multi-agent scenarios. This work serves as an exploratory study on continuous-time multi-agent reinforcement learning problems – a domain where critical issues remain unexplored. We will specify this limitation in the final draft and try to address it in future research.​
>
> [1] R. S. Sutton, C. Szepesv́ari, and H. R. Maei. A convergent O(n) temporal-difference algorithm for off-policy learning with linear function approximation. In NIPS, 2008.

---

> > ### Comment · Reviewer_VtE3 · 2025-08-03
> >
> > Most of my concerns have been addressed. I will keep my score. Thanks!

---

### Official Review · Reviewer_8xLG · 2025-07-02

**Clarity:** 3
**Significance:** 4
**Originality:** 4
**Rating:** 5
**Confidence:** 3

**Summary:**

This paper proposes Continuous Soft Actor-Critic (CSAC), a novel off-policy reinforcement learning algorithm designed for stochastic continuous-time environments. The authors address a fundamental limitation of existing deep reinforcement learning algorithms: their sensitivity to time discretization parameters, which reduces performance in real-world scenarios where small time steps are required. The key innovation lies in using martingale orthogonality conditions from stochastic control theory for policy evaluation, replacing the traditional mean squared temporal difference error approach that fails in stochastic continuous settings. The framework is extended to multi-agent systems (CMASAC) and includes theoretical scaling laws for hyperparameters as time discretization changes. Experimental validation on  benchmarks demonstrates improved robustness compared to existing discrete time algorithms

**Questions:**

Is it possible to implement  the methods laid out in the paper in more traditional benchmarks such as MujoCo and OpenAI gym? Is there a computational constraint in order to do that?

**Ethical Concerns:**

["NO or VERY MINOR ethics concerns only"]

**Final Justification:**

I have gone over the replies of the reviewers. They have answered the questions to my satisfaction, and I maintain my positive score.

**Limitations:**

The only limitation that I can think of is the method not applied to traditional benchmarks such as MujoCo and OpenAI gym, although it is reasonable to think this might be a future work.

**Quality:**

4

**Strengths And Weaknesses:**

Strengths
1. The paper proposes a new off-policy actor-critic algorithm, Continuous Soft Actor-Critic (CSAC), that operates in a stochastic continuous-time setting. This addresses a crucial gap in reinforcement learning literature, where time discretization often causes performance degradation.

2. The extension to continuous-time multi-agent settings (CMASAC) with a centralized training/decentralized execution (CTDE) scheme is well-motivated and executed, supporting both cooperative and competitive environments.

3. Experiments on VMAS tasks show that continuous time methods outperform baselines in small time increment regimes, demonstrating practical robustness. The inclusion of hyperparameter scaling laws for discretization adds substantial value for real-world deployment.


Weakness.
1. While the paper is technically rigorous, it may be difficult for readers without a strong background in stochastic control to follow. The main text could better motivate some of the theoretical formulations with more intuitive explanations.

---

> ### Author Rebuttal · Authors · 2025-07-29
>
> Thank you for your thoughtful comments. We sincerely appreciate the reviewer's approval of our research problem and the proposed methodology. Our responses to the inquiries are as follows:
>
> 1. To improve readability and clarity of the paper, we will expand the explanation of conclusions in the final draft. In particular, we will clarify that minimizing MSTDE to zero will not lead to the learned parameters converge to their true values (in continuous stochastic setting, the MSTDE corresponding to the true value function is not zero). We will also discuss the use and interpretation of the behavior policy $\tilde{\pi}$ and the target policy $\pi$ in Theorem 1.
>
> 2. Regarding whether the proposed method can be applied to a broader range of benchmarks (e.g., MuJoCo and OpenAI Gym), we agree that the wider the testing, the better. Although we have not yet explored all aspects in the current study, our method does not rely on specific information from the system dynamic. Therefore, we believe that the method and its implementation can be tested in any simulation environment with small time-scale dynamics. As shown in Section 2.1, the method proposed in this paper is discussed within the SDE framework, and therefore cannot guarantee excellent performance in all scenarios. In practice, we usually only have access to data and cannot get the statistical characteristics of the noise (e.g., white noise? independent increments? Markovian?). Thus, for problems with small time discretization (especially in cases that employ a Gaussian distribution policy like this paper), the method proposed in this paper should be included in the comparative analysis. We have added an additional single-agent experiment, as well as the multi-agent comparison experiments with more random seeds and hyperparameter settings (once the update is approved by the official, we will release the additional experimental results).

---

### Official Review · Reviewer_xz8q · 2025-07-02

**Clarity:** 3
**Significance:** 3
**Originality:** 3
**Rating:** 4
**Confidence:** 4

**Summary:**

**Strengths:**

The authors present an off-policy value function approximation result for continuous time reinforcement learning (RL). They also provide algorithms for policy gradients and multi agent RL. They prove under what kind of time discretization and hyper-parameter scheme their algorithm parameters converge or diverge. They demonstrate the success of time scaling with two different time-discretizations.

**Weaknesses:**

 See questions below.

**Questions:**

**Major issues**
================

**Section 3.1:**
I believe you can motivate why the MSTDE is not sufficient a bit better. I am unclear as to how equation 9 shows that "minimizing MSTDE to learn parameterized function J^(\theta) cannot guarantee convergence to value function J". I believe Section 3 in Jia and Zhou [2022a] elaborates a lot more on this issue. Further, in statement of Theorem 1 I am unclear whether you are stating that the value function approximation J^(\theta) obeys eqn 11 or the true value function obeys this? Presumably both one as a matter of fact and the other as a constraint you optimize towards? The source of this confusion is from the proof and in the equation below line 398 you introduce J^(\theta) into the equation without sufficient justification. You describe the approximating function $J^{\theta}$ on line 143 but I am not sure how it shows up in the proof of Theorem 1 under line 398 and also under line 400. How does the assumption over the polynomial growth in $\theta$ of the partials $\partial J(\theta)/\partial θ, \partial^2J(\theta)/\partial \theta^2$ factor into this theorem?

**Question on the probability spaces you use:** On lines 364-365 you state that stochastic optimal control problems’ probability space is $(\Omega, \mathcal F, \mathbb P^W)$. I believe, you utilise this fact by using the same $ \omega$ to fix the state ($X_s^{\tilde{\pi}}$), the action/control ($u_s^{\tilde{\pi}}$) and the exit time. I am unsure how you are able to define the stochastic processes associated with the state and the control over the same probability space? This raises the question of application of Fubini’s theorem in the series of equations below line 392. If the definition over the same probability space is in fact correct then we are good but in the case that is not true one cannot apply the Fubini’s theorem (as you might well be aware). Could the authors please justify this a bit more?

**Proof of Theorem 1:** Could you please comment on where the off-policy nature of this result is ensured or introduced in the proof? Equation 22 and Equation equation below line 398 both do not have the policy in them.

**Proof of Theorem 3:** How do you obtain the approximation $\approx$ in equation 33? How do you go from $\partial J/\partial t$ to $d J$ in this equation? Please let me know if you have defined this somewhere in the text.

**Question on MADDPG:** How do you adapt DDPG (with re-parameterized value functions) for the continuous time setting? As far as I can understand, the results (theorem 1 and 3) are for policies that sample from some Gaussian distribution. In the DDPG (with reparameterized policy gradients) the functional form of policy need not match what we see in the result of theorem 3.


----------------
**Minor issues:**
====================

Line 93: n-dimensiona → n-dimensional
Line 96: can you please explain why you have this additional expression for filtration and what $\sigma$ is here?
Section 3.1: You mention that problems 1-2 and 3-4 remain well-posed throughout the analysis → can you please point to the list of conditions for this in the Appendix? This is for the sake of completeness.

Line 123: is it not obvious to me how $M_s$ is a square-integrable martingale. Could you explain this please. I am also not super clear on the definition of J in equation 6: what is $t’$? Further, in the definition of $M_s$ is there also a dependence on $t$ or $\pi$?

Equation 12 and Theorem 1: Can you intuitively explain the difference between $\pi$ and $\tilde{\pi}$ before introducing them as symbols please? (I would recommend moving the explanation from line 147).

Equation 13: what is $\pi^{\phi}$?
At some places you denote $J$ with $J (\cdot; \pi)$ (e.g. theorem 1) and in other places it is just $J(\cdot)$ (e.g. equation 9). This is a bit confusing for me. Also, the same issue under line 446.

Assumption 1 → could you please expand on the “polynomial growth condition in x” in the main body? I see the description in section A.2 but it might be helpful to define it in the main body where you state this assumption.

Section A.2 (and the assumptions 2 and Definition 1) is not referenced while stating theorem 3 in the main body. I believe this should be done somewhere in the main body for readers to get a complete picture. I have not cross referenced the assumptions with Jia and Zhou but I believe you have stated all of them in your work?

Line 375: Can you please support in the text how the process in eqn 20 is a “continuous local martingale with finite variation”?

Line 367: Where is $\pi$ in the equation below line 367?

Section 3.1: You initially present the cumulative reward setting and follow it up with a result in the discounted reward setting (Theorem 3).
I am curious if all the assertions (square-integrable martingale, MSTDE etc) also apply for the discounted setting? Or are these results not applicable to the discounted setting?

Line 170: Let J is the value function with policy → Let J be the value function with policy
Line 169: what is the broader setting here? off-policy policy gradient?

Line 401-402: Could you please give a pointer or reference to viscosity solutions here.

**Ethical Concerns:**

["NO or VERY MINOR ethics concerns only"]

**Final Justification:**

I had five major issues. All of these have been resolved by the authors and I hope these changes can make it to the final version if the paper is accepted. they have also provided citations for some key missing elements. I believe this is original theoretical work that works towards understanding multi agent reinforcement learning and it would be a good contribution to the conference. Despite other reviewers pointing towards limitations in empirical studies, I think the contributions are significant.

**Limitations:**

See major issues above. In summary, a few key points need more clarification (via references or citations) and the results need to be explained a bit better.

**Paper Formatting Concerns:**

None.

**Quality:**

3

**Strengths And Weaknesses:**

Strengths:

The authors build the problem and the solution up fairly carefully for the reader. they provide essential results for policy gradients and multi agent RL in continuous time RL. They have experiments with existing algorithms and environments.

---

> ### Author Rebuttal · Authors · 2025-07-29
>
> We sincerely appreciate your valuable comments. We found them helpful in improving our draft. Please find our responses to your questions below.
>
> Answers for Question in Section 3.1:
>
> 1. We hope that the design of the loss function can reflect the function approximation error, whereas the quadratic variation reflects the noise or fluctuations in the process. A reduction in the expected quadratic variation of a process can only suppress volatility, but does not guarantee that $ J^{\theta} $ is closer to the true function $ J $. Or conversely, if we use MSTDE as the loss function, the learned result will be a set of parameters that make the MSTDE equal to zero. However, the MSTDE corresponding to the true value function (martingale) is not zero (Equation (9)).
>
> 2. Equation (11) is a process that we define, and it can be defined for any given function $J$, whether it is an approximation function or the true value function. We need to examine whether the defined process is a martingale. This determines whether the given function $ J $ is indeed the value function with $\pi$. If we can ensure that the process $ M_s $ is a martingale (by using data generated by an arbitrary behavior policy $\tilde{\pi}$), then $ J $ is the true value function. In the proof of Theorem 1, by the martingale property, we have shown that Equation (21) holds. The validity of Equation (21) ensures that the equation below line 398 holds. If we take the arbitrary function given in Theorem 1 to be $ J^{\theta} $, then Equation (22) follows. In addition, Equation (22) is a sufficient condition for $ J $ to be the value function associated with the policy $ \pi $. The assumption on partial derivative with respect to $\theta$ does not appear in proof of Theorem 1. However, it is good for stability and convergence during the update process of $\theta$ (see Sutton [2008] and Jia and Zhou [2022b], same with Assumption 2).
>
> Answers for Question on the probability spaces:
>
> This is feasible. In this paper, the stochasticity originates from two mutually independent sources: Brownian motion (describing environmental randomness) and randomness from the policy. We can construct a sample space that accommodates them. If your concern is regarding the measurability issue of continuum independent sampling in a continuous framework. In response, we wish to cite references Jia and Zhou [2025] and Sun [2006](in the final of the rebuttal), which provide detailed and in-depth explorations of this issue. However, in the context of this paper, this issue does not impact our results and methods. Our framework and conclusions are built on continuity, but our goal is to support viable algorithms (real-world implementation always occurs in a discrete setting). Considering the focus and readability of the article, we did not delve into this detail in the text. We appreciate the reviewer's questions and comments, and we will add a note regarding this point in the final version of our paper.
>
> Answers for Proof of Theorem 1:
>
> Being off-policy means that the transition data used in equations (11) and (19) originates from an arbitrary behavior policy $\tilde{\pi}$. Theorem 1 demonstrates that, as long as the martingale condition is satisfied, we are able to get the value function $J$ of the target policy $\pi$. Consequently, the content of line 398 pertains to the target policy $\pi$. The role of $\tilde{\pi}$ is solely to characterize the data-generating process. Roughly speaking, one can view the problem introduced by off-policy data in previous approaches as having been transformed into ensuring the martingale condition in Equation (11).
>
> Answers for Proof of Theorem 3:
>
> The approximation in Equation (33) comes from Line 453.
>
> Answers for Question on MADDPG:
>
> Adapting DDPG to the continuous-time setting is a valuable question. In this paper, we do not address this issue, and it will be considered as one of our future research directions. In the present work, we only use the MADDPG method from Lowe et al. [2017] and Bettini et al. [2024] as baseline algorithms to evaluate the performance of our proposed approach.
>
> Answers for Minor issues:
>
> Line 93 and Line 170: We appreciate you for pointing that out. We will make the correction.
>
> Line 96: In this paper, the state and action are interdependent. The randomness in the agent-environment interaction comes from two sources: one is the Brownian motion representing the environment, and the other is of the policy itself. Therefore, we define a filtration as shown in Line 97. $\sigma$ is the sigma algebra generated by process $Z_t$.
>
> Section 3.1: Assumption 2 along with Definition 1 in Section A.2 (in the Appendix) can guarantee the well-poseness.
>
> Line 123: When $J$ is the true value function, the Bellman equation implies that the process $M_s$ satisfies the definition of the martingale : $E[M_s|F_t]=M_t$, and is therefore a martingale. In equation (6), time $t^\prime \in [t,T]$. The definition of $M_s$ (11) is dependent on time $s$ and policy $\pi$.
>
> Equation 12 and Theorem 1: $\pi$ is target policy and $\tilde{\pi}$ is behaviour policy.
>
> Equation 13: $\pi^\phi$ is a parameterized policy, where $\phi$ denotes the parameters of the policy neural network in this paper. In Theorem 1, $J$ is the value function corresponding to the policy $\pi$. To distinguish it from $\tilde{\pi}$, we explicitly denote it.
>
> Assumption 1 and Section A.2: Thank you for your valuable suggestion. We will revise the corresponding part in the main body to ensure the assumptions are comprehensive and the content is reader-friendly.
>
> Line 375: This follows from Assumptions 1 and 2 in this paper, which ensure continuity. Moreover, as it is a Lebesgue integral, it necessarily has finite variation.
>
> Line 367: The formula under line 367 is not the HJB equation corresponding to the stochastic policy $\pi$, hence it does not include the policy $\pi$.​​
>
> Section 3.1: Yes, these discussions apply to the discounted setting. Their occurrence arises from Equation (32)(Theorem 3). And the assertions (square-integrable martingale, MSTDE etc) also apply for the discounted setting. They arise primarily due to continuous stochastic environment (noise satisfies the statistical properties of Brownian motion) rather than discounting.
>
> Line 169:  This is the policy gradient of the strategy $\pi^\phi$, and therefore it is on-policy. We will clarify this in the final draft to avoid confusion.
>
> Line 401-402: Yes, we hereby list the references Fleming [2006] and Beck [2021].
>
> [1] C. Beck, M. Hutzenthaler, and A. Jentzen. On nonlinear Feynman–Kac formulas for viscosity solutions of semilinear parabolic partial differential equations. Stochastics and Dynamics, 2021.
>
> [2] W. H. Fleming and H. M. Soner. Controlled Markov Processes and Viscosity Solutions, volume 25. Springer Science \& Business Media, 2006.
>
> In addition to the above points, to more intuitively illustrate the differences between our method (off-policy) with methods proposed by Jia and Zhou [2022, 2023](on policy), we conduct two sets of comparative experiments using the linear-quadratic example from Jia and Zhou [2023, Section 7] and the code provided by them. We set the behavior policies as normal distributions with mean parameters of (0.5, 0.5) and (1, 1), respectively, and a variance of 1 for both. We observe that the method from Jia and Zhou [2022, 2023] collapse (oscillatory collapse and significant deviation from the true value), while our approach remains stable (once the update is approved by the official, we will release the results). We simply list the learning error ($\phi_2$-$\phi_2^{true}$) of the parameter $\phi_2$ when the behaviour policy is set to $N(x + 1, 1)$.
>
> |iteration(10^4)|2|4|6|8|10|
> |------------|------------------|------------------|------------------|------------------|------------------|
> |CSAC| 0.706|0.780|0.701|0.683|0.670|
> |q learning(Jia and Zhou,2023)|-0.586|-4.211|5.756|-4.053|-3.664|
> |continuous TD(0)(Jia and Zhou,2022)|-4.149|-4.142|-4.139|-4.112|-4.098|
>
> [3] R. S. Sutton, C. Szepesv́ari, and H. R. Maei. A convergent {O}(n) temporal-difference algorithm for off-policy learning with linear function approximation. In NIPS, 2008.
>
> [4] Y. W. Jia and X. Y. Zhou. Erratum to "q-Learning in Continuous Time", 2025.
>
> [5] Y. N. Sun. The exact law of large numbers via Fubini extension and characterization of insurable risks. Journal of Economic Theory, 126, 1,31-69,2006.
>
> We hope this response has clarified your question.

---

> > ### Comment · Reviewer_xz8q · 2025-08-04
> > **Response to the authors**
> >
> > thank you for your detailed response to all my concerns.
> >
> > On the probability measure issue: I took a quick look at Jia and Zhou, 2025, I believe Jia and Zhou, 2025 are defining a distinct probability space (equation 1) for time-discretized samples of actions: $( \Omega, \mathcal F, \mathbb P)$. Albeit the this discretization is over any grid. This leads to them defining the action process, $a_s$, adapted to $\mathcal F_s$. They then derive the results from Jia and Zhou (2023) for any time grid $\mathscr G$. This gives me the impression that defining a probability space to whose filtrations all these different processes are adapted requires some explicit work. How do you see your proofs changing? Are there any qualitative or essential changes if you were to follow this approach?
> >
> > Line 96: for the generated sigma algebra I would use a different symbol because it is being reused for the diffusion term in eqn 1. I understand this is more common in probability theory literature but it might be helpful to make the distinction.
> >
> >
> > Y. W. Jia and X. Y. Zhou. Erratum to "q-Learning in Continuous Time", 2025.

---

> > > ### Author Response · Authors · 2025-08-05
> > >
> > > 1.We sincerely appreciate your response. Regarding this issue, we believe this stems from different choices in problem formulation. As is well known, in the field of reinforcement learning, execution and information exchange​​ necessarily occur in discrete time (as clarified in Section 2, where $x_t$ and $u_t$ correspond to the agent's state and generated actions). Therefore, the potential issues associated with the "technical gap" in the framework of continuous-time stochastic analysis (i.e., measurability and the well-posedness of integrals on extended probability spaces) do not actually arise. If we set the space to the representation form of Jia and Zhou [2025, Equation (1)] (i.e., by considering discrete-time action processes), it would not affect the proofs and conclusions of this paper. Following Jia and Zhou [2025], we would only need to replace the time index of the action process with a discrete format (along with the corresponding filtration). If the framework adopts a discrete representation—where the measure space, system dynamics (state, action, and reward processes), and martingale conditions are all formulated in discrete time—it would align more closely with practical implementations. However, this would increase the complexity of notation, conceptualization, and comprehension, which is one of the reasons why we employ a continuous-time formulation.
> > >
> > > 2.We are truly sorry for misunderstanding your valuable comment. To address this, we will make a clear distinction of $\sigma$ in the final draft.

---

> > > > ### Comment · Reviewer_xz8q · 2025-08-05
> > > > **Updating my score**
> > > >
> > > > Thank you for your detailed response. It has helped me understand your work and related work better. I believe you can make all these fixes and improve parts of the presentation. I am updating my score to reflect this.

---

### Note · Authors · 2025-08-11

Dear AC and Reviewers,

We appreciate the reviewers for their thoughtful and constructive feedback. This paper investigates the critical issue of sensitivity to time-discretization parameters in RL algorithms, providing results for off-policy RL in continuous-time settings. We are encouraged by the reviewers' recognition of our work’s contributions. We have submitted our rebuttal in response to the reviewers’ concerns and believe we have addressed the majority. Below, we summarize the key responses.

1. Technical Details:
We have further clarified why mean square TD error (MSTDE) fails in certain scenarios and elaborated on potentially ambiguous theoretical details in this paper. This aims to enhance intuitive understanding of our claims for readers without a background in stochastic control.

2. Additional Experiments:
We have added a new single-agent comparative experiment to intuitively illustrate the distinctions between our method and prior work. Additionally, we have included a multi-agent experiment to address concerns about the experimental configuration. The results align with the paper’s conclusions, which helps address concerns.

3. Applicability:
Reviewer concerns centered on the applicability scope of the method proposed in this paper. We believe that the current concentration of the RL community on a few common simulation environments stems partly from insufficient development in algorithmic research and open-source simulation platforms. This paper explicitly targets continuous state spaces; thus, environments like Overcooked fall outside its scope. Modeling continuous stochastic problems with SDEs is a natural approach, and it has played a significant role in applications such as generative models. Gaussian noise is also a common process noise, for example, in wind resistance and sensor systems. The applicability of existing methods under such near-continuous-time noise should be discussed (e.g., MSTDE’s limitations highlighted in the paper). Furthermore, we have provided references illustrating MARL’s applicability to buyers and sellers in financial markets, to clarify the practical utility of the multi-agent method.

We are grateful to the AC and the reviewers for taking the time to review and discuss our paper. Your efforts have been invaluable in clarifying our arguments and improving the quality of this paper.

---

### Decision · Program_Chairs · 2025-09-17

**Decision:**

Accept (poster)

**Comment:**

This paper proposes CSAC, an off-policy RL algorithm for continuous-time stochastic environments using martingale orthogonality conditions instead of traditional MSTDE approaches. Four reviewers provided scores ranging from 4-5, with initial concerns about mathematical novelties, experimental validation, and practical relevance largely addressed through comprehensive author rebuttals. The main strengths include solid theoretical foundations addressing limitations of existing methods in stochastic continuous settings and novel hyperparameter scaling laws, while weaknesses center on limited experimental scope (only VMAS environments) and unclear broader applicability beyond specialized domains like financial markets. The consensus supports borderline acceptance based on meaningful theoretical contributions despite concerns about immediate practical impact.